# Breaking the Reversal Curse: How Masked Diffusion Models Achieve Reverse Inference

## Abstract

The reversal curse, failing to answer "$B$ is $A$" after learning "$A$ is $B$", is a persistent pathology of autoregressive language models (ARMs). Masked diffusion based language models (MDMs), however, appear to escape this curse. A seemingly plausible explanation attributes this ability to their any-order training objective, but we show this intuition is incomplete. In particular, training to replace the mask in "[**M**] is $B$" with $A$ learns the probability $p(x = A|y = B)$, which has nothing to do with the probability required to answer the reverse query, $p(y = A|x = B)$. Thus, the objective formulation alone cannot explain reversal ability. We demonstrate that the true reason lies in the architecture: in a one-layer Transformer encoder, attention scores for forward and reverse contexts are positively correlated, implicitly coupling probabilities that would otherwise be treated as unrelated. This structural bias gives MDMs a principled advantage for reverse inference. Our theory is supported by both synthetic and real-world experiments, where MDMs consistently succeed on reverse queries that cause even strong ARMs to fail.

## 1 Introduction

Since the advent of the Transformer architecture (Vaswani et al., 2017), language models have advanced rapidly (Devlin et al., 2019; Raffel et al., 2020). Autoregressive Models (ARMs) (Radford et al., 2018; 2019; Brown et al., 2020), implemented as Transformer decoders and trained with next-token prediction, have become the dominant paradigm for large language models (LLMs) (Grattafiori et al., 2024; OpenAI, 2023). Despite their success, ARMs exhibit structural limitations. A notable example is the *reversal curse* (Berglund et al., 2024): after learning the fact "$A$ is $B$", they often fail to answer the logically equivalent reverse query "$B$ is $A$". This arises because ARMs are optimized only for the unidirectional conditional probability $p(y = B|x = A)$, without explicitly modeling the reverse probability $p(y = A|x = B)$. For instance, a model may correctly predict "The capital of France is Paris," yet fail to answer "Which country has Paris as its capital?" Data augmentation techniques (Golovneva et al., 2024; Lu et al., 2024; Lv et al., 2024; Zhang et al., 2025) can partially alleviate the problem, but do not resolve the bias fundamentally.

Masked diffusion based language models (MDMs) (Austin et al., 2021; Campbell et al., 2022; Lou et al., 2024; Sahoo et al., 2024; Shi et al., 2024; Ou et al., 2025), implemented with Transformer encoders and trained via random masking and reconstruction, have recently emerged as a promising alternative to ARMs. They offer several advantages: the encoder architecture naturally supports bidirectional context modeling, the random masking objective enables generation in any order, and recent work has demonstrated their scalability to the LLM regime (Nie et al., 2025b; Ye et al., 2025). In addition, MDMs have been reported to handle reverse queries more effectively than ARMs (Kitouni et al., 2024; Nie et al., 2025a;b), suggesting a potential structural advantage. However, these observations remain anecdotal, and no systematic analysis has yet been provided.

We begin by establishing, through systematic experiments on large-scale language models, that MDMs indeed mitigate the reversal curse. Whereas prior work focused only on smaller models at the 1.1B scale (Nie et al., 2025a), we conduct controlled evaluations at the 7–8B scale, comparing ARMs (LLaMA-3.1 (Grattafiori et al., 2024), Qwen-2.5 (Yang et al., 2025)) with an MDM (LLaDA (Nie et al., 2025b)). Across real-world benchmarks such as Parent–Child and Person–Description, we find that MDMs consistently succeed on reverse inference tasks where strong ARMs collapse. These

large-scale results provide the first systematic evidence that the reversal curse is substantially alleviated in MDMs under realistic evaluation settings.

Having established the phenomenon, we then ask: *why do MDMs succeed where ARMs fail?* A common intuition points to the any-order nature of the MDM training objective: random masking provides supervision across all conditional directions. Yet, this explanation is incomplete. By formulation, the probability of unmasking [**M**] as $A$ in "[**M**] is $B$" corresponds to $p(x = A|y = B)$, whereas the reverse query "$B$ is [**M**]" requires $p(y = A|x = B)$. These two conditional probabilities are defined with respect to different conditioning events, and the training objective provides no mechanism to establish a systematic relation between them.

We demonstrate that the key to reversal ability lies in the Transformer encoder architecture of MDMs. Under a simplified setting of a one-layer encoder, we provide a formal proof that the attention score reinforced during forward training is positively correlated with the attention score required for reverse inference. This architectural property couples conditionals that are otherwise unrelated, giving MDMs an inherent advantage for reversal. A controlled toy experiment further confirms this effect, showing that the theoretical prediction aligns with empirical behavior and complements our large-scale findings.

In summary, our contributions are:

- **Large-scale experiments:** We systematically evaluate 7–8B parameter models and show that MDMs consistently outperform ARMs on reversal tasks.

- **Theoretical insight:** We prove that reversal ability in MDMs comes from the Transformer encoder architecture, where attention scores for restoring "$A$" from "[**M**] is $B$" and from "$B$ is [**M**]" are positively correlated.

- **Empirical validation:** Synthetic toy experiments confirm the theoretical prediction and align with our large-scale results.

## 2 PRELIMINARIES

### 2.1 AUTOREGRESSIVE MODELS AND MASKED DIFFUSION MODELS

In this section, we review autoregressive models (ARMs) and masked diffusion models (MDMs) with a focus on their training objectives and architectures. Within the architecture, our analysis centers on the self-attention mechanism of the Transformer encoder used in MDMs, which governs how models process context and is crucial for understanding their capacity for reverse inference.

**Training Objectives.** An ARM (Radford et al., 2018; 2019) is trained to generate a sequence $\boldsymbol{x} = x_1 x_2 \ldots x_L$ strictly in a left-to-right manner. Given a prefix $\boldsymbol{x}_{<i} = x_1 x_2 \ldots x_{i-1}$, the model maximizes the conditional probability of the next token $x_i$. Formally, the training objective is the following cross-entropy loss:

$$\mathcal{L}_{\text{ARM}}(\theta) = -\mathbb{E}_{\boldsymbol{x} \sim p_{\text{data}}} \left[ \sum_{i=1}^{L} \log p_\theta(x_i | \boldsymbol{x}_{<i}) \right].$$

By contrast, an MDM (Sahoo et al., 2024; Shi et al., 2024; Ou et al., 2025) learns to generate a sequence in an any-order fashion via random masking. Let $\boldsymbol{x}^t$ denote a corrupted version of $\boldsymbol{x}$ in which each token is independently replaced by the special mask token [**M**] with probability $t \in [0, 1]$. The model is then trained to recover the original tokens at the masked positions by maximizing the conditional probability of each masked token. The formal training objective is the following weighted cross-entropy loss:

$$\mathcal{L}_{\text{MDM}}(\theta) = -\mathbb{E}_{\boldsymbol{x} \sim p_{\text{data}}, \, t \sim \mathcal{U}[0,1], \, \boldsymbol{x}^t} \left[ \frac{1}{t} \sum_{i:x_i^t = [\mathbf{M}]} \log p_\theta(x_i | \boldsymbol{x}_{\text{UM}}^t) \right],$$

where $\boldsymbol{x}_{\text{UM}}^t$ denotes the unmasked portion of $\boldsymbol{x}^t$.

**Architectures.** An ARM models $p_\theta(x_i|\boldsymbol{x}_{<i})$ with a Transformer decoder that uses causal attention. At each step $i$, the decoder takes the prefix $\boldsymbol{x}_{<i}$ as input and produces a probability distribution over the vocabulary $\mathcal{V}$, from which the next token $x_i$ is drawn.

In contrast, an MDM models $p_\theta(x_i|\boldsymbol{x}_{\text{UM}}^t)$ with a Transformer encoder that applies full-attention. The encoder processes the corrupted sequence $\boldsymbol{x}^t$, which contains $[\mathbf{M}]$ at a subset of positions, and produces a distribution over $\mathcal{V}$ at every position $i$. Only the outputs at masked positions are meaningful, as they specify the probabilities of reconstructing the masked tokens.

**Self-Attention in the Transformer Encoder.** A central component of MDMs is the self-attention mechanism in the Transformer encoder, which governs how information flows across tokens in a sequence. Since our theoretical analysis hinges on this mechanism, we describe it carefully in the single-head case with head dimension $D$.

Each input token embedding $\mathbf{h}_i \in \mathbb{R}^D$ is projected into a query, key, and value vector via shared projection matrices $\mathbf{W_Q}, \mathbf{W_K}, \mathbf{W_V} \in \mathbb{R}^{D \times D}$:

$$\mathbf{q}_i = \mathbf{W_Q}\mathbf{h}_i, \quad \mathbf{k}_i = \mathbf{W_K}\mathbf{h}_i, \quad \mathbf{v}_i = \mathbf{W_V}\mathbf{h}_i.$$

The interaction between token $i$ and token $j$ is first measured by an *attention score*. This score captures how strongly the query at position $i$ attends to the key at position $j$, combining semantic similarity (through the projections) with relative positional information introduced by Rotary Position Embedding (RoPE) (Su et al., 2024):

$$\text{Score}(i, j) \;=\; \mathbf{q}_i^\top \, \mathbf{R}(\Delta) \, \mathbf{k}_j,$$

where $\mathbf{R}(\Delta) \in \mathbb{R}^{D \times D}$ is the RoPE matrix determined by the relative position $\Delta = j - i$.

Raw scores are normalized with softmax to produce *attention weights*:

$$\text{Weight}(i, j) \;=\; \frac{\exp\left(\frac{1}{\sqrt{D}}\text{Score}(i, j)\right)}{\sum_{j'=1}^{L} \exp\left(\frac{1}{\sqrt{D}}\text{Score}(i, j')\right)}.$$

The attention weight represents how much token $i$ focuses on token $j$. In other words, it determines how much the representation at position $i$ will incorporate information coming from position $j$.

The output at position $i$, the *context vector*, is then a weighted combination of value vectors:

$$\mathbf{z}_i \;=\; \sum_{j=1}^{L} \text{Weight}(i, j) \, \mathbf{v}_j.$$

In practice, whether token $i$ relies on token $j$ (for example, whether a masked token $[\mathbf{M}]$ attends to $B$ to reconstruct $A$) is entirely governed by this attention distribution. This mechanism, which couples forward and reverse contexts, is central to our theoretical analysis.

## 2.2 The Reversal Curse in Autoregressive Models

As discussed in Section 2.1, autoregressive models (ARMs) generate text in a strictly left-to-right manner. This design leads to the well-documented *reversal curse* (Berglund et al., 2024): even after learning the forward relation "$A$ is $B$," ARMs frequently fail to answer the logically equivalent reverse query "$B$ is $A$." For instance, a model may correctly predict "The capital of France is Paris," yet fail to respond to "Which country has Paris as its capital?"

Lin et al. (2024) provided a broad examination of this phenomenon across open-ended QA and multiple-choice settings. They showed that ARMs succeed on reversed queries only when both entities are explicitly present in context, and identified a name-centric "thinking bias" that ties generalization ability to the structural form of training data.

Several approaches have attempted to mitigate the reversal curse through data-centric interventions. Golovneva et al. (2024) proposed reverse training, augmenting pre-training or fine-tuning with reversed variants of each sequence (token-level, word-level, entity-preserving, or random-segment reversals) under the same left-to-right objective. They reported that entity-preserving and random-segment reversals substantially reduce the reversal curse without degrading performance on standard

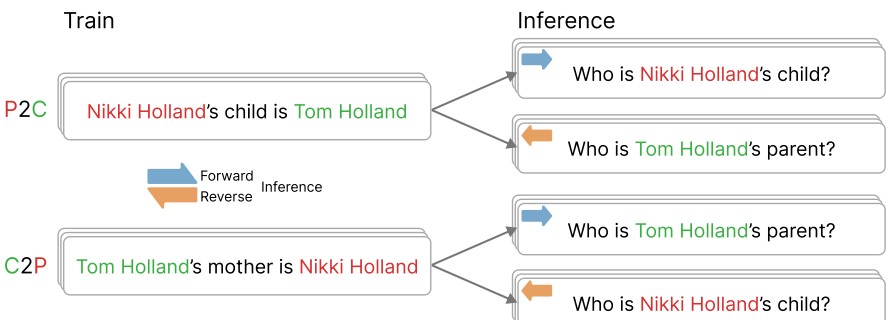

Figure 1: Illustration of the evaluation setup on the Parent–Child dataset. Each model is trained only in one direction (e.g., parent→child or child→parent) and then evaluated on both forward and reverse queries. The figure highlights representative prompts and completions, where forward queries follow the trained mapping and reverse queries require the unseen inverse mapping. Exact-match accuracy on such queries quantifies reverse inference ability. The Person–Description dataset follows the same setup.

benchmarks, and highlighted the importance of augmentation granularity. Complementary to this, Lu et al. (2024) analyzed three contributing factors (knowledge clarity, entity-correlation modeling, and pairwise reasoning) and quantified their effects via controlled experiments.

Despite these efforts, a fundamental limitation remains. An ARM learns the relation "$A$ is $B$" by maximizing only the forward conditional probability $p_\theta(y = B|x = A)$. This training objective is entirely decoupled from the reverse conditional $p_\theta(y = A|x = B)$, making reverse inference an independent task rather than a byproduct of forward learning. This limitation is not just intuitive but also formal: a gradient-based analysis of a one-layer Transformer decoder shows that optimizing $p_\theta(y = B|x = A)$ provides no signal for improving $p_\theta(y = A|x = B)$ (Zhu et al., 2024). Although this analysis is carried out in the decoder setting, the difficulty stems from the next-token prediction objective itself, which makes reverse inference intrinsically hard to achieve.

## 3 LARGE-SCALE SYSTEMATIC EXPERIMENTAL ANALYSIS

As discussed in Section 2.2, autoregressive next-token prediction optimizes a single directional conditional, which prevents ARMs from answering reversal queries. By contrast, MDMs receive bidirectional supervision via random masking and have been reported to alleviate the reversal curse (Kitouni et al., 2024; Nie et al., 2025a;b). We provide the first large-scale and systematic experimental comparison of ARMs and MDMs on reverse inference.

**Setup.** Training data includes only forward statements of the form "$A$ is $B$," while the reversed form "$B$ is $A$" is never provided. At evaluation, we test both directions:

- **Forward ("$A$ is $B$"):** given "$A$ is _," predict the next token $B$ (ARM);
  given "$A$ is [**M**]," predict the masked token $B$ (MDM).

- **Reverse ("$B$ is $A$"):** given "$B$ is _," predict the next token $A$ (ARM);
  given "$B$ is [**M**]," predict the masked token $A$ (MDM).

We use three real-world tasks adapted from Berglund et al. (2024) and Elsahar et al. (2018): *Parent–Child, Person–Description* and *T-REx*. For the T-REx task, we select six relations from the original T-REx triplets and convert each entity into a virtual name to generate natural sentences used for training. All tasks provide unambiguous mappings between entities, where forward queries match the training direction and reverse queries swap input and output. Figure 1 illustrates representative forward and reverse examples. We report exact-match accuracy after minimal normalization, with further dataset details provided in Appendix D.

Table 1: Results of Parent–Child, Person–Description and T-REx datasets for real-world evaluation. Train Dataset indicates the direction of data used for training. **Across all cases, LLaDA (MDM) shows notably strong performance in Reverse accuracy.** The highest Reverse accuracy for each training direction is boldfaced, all achieved by LLaDA. In contrast, LLaMA-3.1 and Qwen-2.5 (ARMs) nearly collapse to random guessing and almost completely fail to perform reverse inference. Results are averaged across 3 random seeds.

| | MDM | | ARM | | | |
| | LLaDA 8B | | LLaMA-3.1 8B | | Qwen-2.5 7B | |
| Train Dataset | Forward | Reverse | Forward | Reverse | Forward | Reverse |
|---|---|---|---|---|---|---|
| Parent → Child (P2C) | 76.7 | **48.3** | 89.9 | 15.9 | 89.9 | 0.5 |
| Child → Parent (C2P) | 87.7 | **43.7** | 95.9 | 6.9 | 89.0 | 1.4 |
| Person → Description (P2D) | 72.7 | **99.5** | 72.7 | 3.5 | 70.7 | 2.2 |
| Description → Person (D2P) | 99.7 | **41.3** | 83.0 | 1.8 | 80.0 | 1.5 |
| T-REx | 92.3 | **81.5** | 87.3 | 2.8 | 89.8 | 2.3 |

**Models.** We evaluate three large-scale LLMs. LLaDA 8B Instruct (Nie et al., 2025b) is a diffusion-based language model that scales MDM to 8B parameters and was developed with LLaMA-3 as its primary comparison target. For ARMs, we include LLaMA-3.1 8B Instruct (Grattafiori et al., 2024) and Qwen-2.5 7B Instruct (Yang et al., 2025). All models are fine-tuned on the same training data using LoRA (Hu et al., 2022), and evaluated with deterministic decoding to ensure consistency.

**Results.** Table 1 reports accuracy on the Parent–Child, Person–Description and T-REx datasets. Both ARMs and the MDM achieve high accuracy in the *Forward* regime, confirming that all models can reliably learn the observed mappings from training data. However, a stark contrast emerges in the *Reverse* regime: LLaMA-3.1 and Qwen-2.5 almost collapse to random guessing, demonstrating the autoregressive reversal curse described in Section 2.2. In sharp contrast, LLaDA consistently achieves strong reverse accuracy across all tasks, despite never being trained on reversed pairs. These results provide systematic large-scale evidence that the reversal curse is substantially alleviated in MDMs, while it persists in ARMs even at billions of parameters.

## 4 WHY MDMs SUCCEED AT REVERSAL

### 4.1 TRAINING OBJECTIVE ALONE DOES NOT EXPLAIN REVERSAL

In Section 3, we showed empirically that MDMs succeed at reverse inference, whereas ARMs fail. A common explanation, repeated explicitly or implicitly in prior work (Kitouni et al., 2024; Nie et al., 2025a;b), is that the random masking objective of MDMs naturally equips them with reversal ability. The reasoning is that for a sequence "$A$ is $B$," the model is trained on both $p_\theta(y = B | x = A)$ from the corrupted sequence "$A$ is [$\mathbf{M}$]," and $p_\theta(x = A | y = B)$ from "[$\mathbf{M}$] is $B$." Since training covers these two directions, one might conclude that the model implicitly learns to handle the reverse query.

This intuition, however, is incomplete. The reverse query "$B$ is [$\mathbf{M}$]" requires

$$p_\theta(y = A | x = B),$$

which is not directly supervised by the training objective. Importantly, $p_\theta(y = A | x = B)$ (the probability needed for reversal) differs from $p_\theta(x = A | y = B)$, which is observed during training. The two conditionals do not have a guaranteed mathematical connection, and training on "$A$ is $B$" alone does not ensure that information transfers between them (see Fig. 2).

This distinction is important: MDM training directly supervises the forward conditionals, while the reverse conditional required for reversal is not explicitly covered. This suggests that the common explanation, that MDMs succeed at reversal simply because they reconstruct randomly masked tokens, does not fully account for the phenomenon.

Consequently, the strong reversal performance observed in practice (Section 3) is unlikely to be explained by the training objective alone. In the following section, we investigate how structural

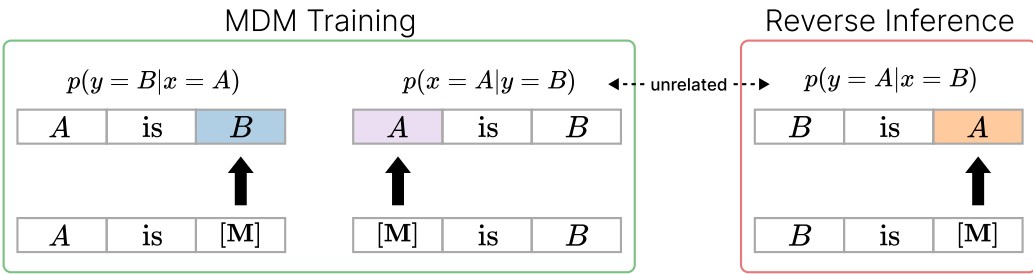

Figure 2: **Why training objective of MDM does not directly enable reverse inference.** When $A$ is masked in "$A$ is $B$," the model only learns to restore $A$ from "[$\mathbf{M}$] is $B$," i.e., $p(x = A|y = B)$. True reversal instead requires $p(y = A|x = B)$, restoring $A$ from "$B$ is [$\mathbf{M}$]." which is mathematically unrelated under the MDM with $p(x = A|y = B)$. Thus, training with random masking cannot by itself explain reversal capability; additional architectural factors must account for the observed success.

properties of the Transformer encoder can implicitly couple forward and reverse attention patterns, providing a more complete explanation for MDMs' reversal capability.

## 4.2 Architecture of MDMs Explains Reversal

As discussed in Section 4.1, the ability of MDMs to perform reverse inference cannot be explained by their training objective. Nevertheless, our experiments in Section 3 showed that once an MDM learns the forward conditional $p_\theta(x = A|y = B)$, it also acquires the reverse conditional $p_\theta(y = A|x = B)$. This raises the key question: *what mechanism in the model couples these otherwise unrelated probabilities?*

We argue that the answer lies in the architecture itself. Specifically, the attention mechanism of the MDM Transformer encoder induces implicit coupling: the attention scores used in forward training are positively correlated with those required for reverse inference. This correlation implies that if the model learns to attend correctly in the forward direction, it will also attend to the right tokens when the order is reversed.

**Setup: One-Layer Transformer Encoder.** We analyze a simplified setting of one-layer Transformer encoder with RoPE, inspired by the analysis of Zhu et al. (2024). In this model, the masked token provides the query vector $\mathbf{q}_{[\mathbf{M}]}$, while each surrounding context token provides a key vector $\mathbf{k}$. The attention score $\mathbf{q}_{[\mathbf{M}]}^\top \mathbf{R}(\Delta)\mathbf{k}$ determines how strongly the masked position attends to a context token. After softmax normalization, these scores yield attention weights, which decide *where the model looks* when unmasking [$\mathbf{M}$].

Reverse inference succeeds if the [$\mathbf{M}$] token attends to the same context tokens it relied on in the forward direction, even when their relative order is swapped. Thus, the central question reduces to whether forward and reverse attention scores are correlated.

**Theoretical Analysis.** As described in Section 2.1, $\mathbf{R}(\Delta)$ denotes the RoPE rotation for relative distance $\Delta$. Consider the forward sequence "[$\mathbf{M}$] is $B$," whose ground truth is "$A$ is $B$." Here the masked token $A$ and the context token $B$ are separated by distance $\Delta_1$, giving the attention score

$$S_{\text{fwd}} = \mathbf{q}_{[\mathbf{M}]}^\top \mathbf{R}(\Delta_1)\mathbf{k}_B.$$

This is the score reinforced during training, since the model must attend to $B$ in order to recover $A$.

In the reversed sequence "$B$ is [$\mathbf{M}$]," the masked token now follows $B$, with relative distance $\Delta_2$. The corresponding attention score is

$$S_{\text{rev}} = \mathbf{q}_{[\mathbf{M}]}^\top \mathbf{R}(-\Delta_2)\mathbf{k}_B,$$

which determines whether the model can again attend to $B$ and correctly infer $A$ in the reverse query.

Although the RoPE rotations differ between the two cases, the key question is whether $S_{\text{fwd}}$ and $S_{\text{rev}}$ move together. Intuitively, if $\mathbf{q}_{[\mathbf{M}]}$ and $\mathbf{k}_B$ align so that the forward score becomes large during training, the rotational structure of RoPE suggests that the reverse score will also tend to be large.

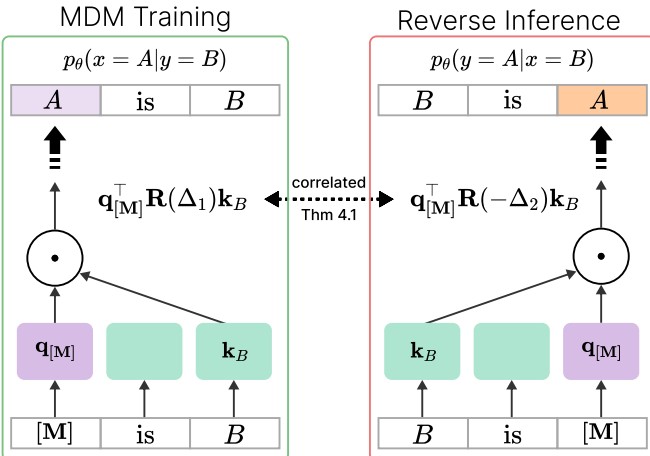

Figure 3: **The mechanism of attention score correlation that enables reverse inference in MDMs.** MDMs are able to infer "$B$ is $A$" although it only learned to reconstruct $A$ from "[**M**] is $B$." i.e., $p_\theta(x = A|y = B)$. For the context "[**M**] is $B$" and the reverse "$B$ is [**M**]", attention scores of [**M**] to $B$ in each context are positively correlated. Induced by the full-attention architecture, the positive correlation associates the two unrelated conditional probabilities (Theorem 4.1). Consequently, the model is able to capture $p_\theta(y = A|x = B)$ and correctly predict "$B$ is $A$" despite never seeing the condition in training.

Formally, we show that forward and reverse attention scores are expected to be positively correlated. We model the query vector $\mathbf{q}$ as an isotropic, zero-mean Gaussian. Given that RoPE operates on disjoint 2D subspaces, we structure the conditional covariance of $\mathbf{k}|\mathbf{q}$ as a block-diagonal matrix with $2 \times 2$ blocks. Within each block, we set the corresponding query subvector $\mathbf{q}_i = (q_{2i-1}, q_{2i})$ as an eigenvector, capturing the expected alignment of $\mathbf{k}_i$ with $\mathbf{q}_i$ after training. Finally, assuming symmetry ($\Delta_1 = \Delta_2 = \Delta$) for analytical simplicity, we obtain the following theorem:

**Theorem 4.1.** *Suppose $\mathbf{q} \sim \mathcal{N}(0, \sigma^2 I_D)$. Let $\mathbf{k} \in \mathbb{R}^D$ be a random vector whose conditional covariance $\text{Cov}(\mathbf{k}|\mathbf{q})$ is block-diagonal with $2 \times 2$ blocks for every realization of $\mathbf{q}$. Assume every block has equal total variance and the $i$-th block admits $\mathbf{q}_i = (q_{2i-1}, q_{2i})$ as an eigenvector. Then, the expected correlation between the forward and reverse attention scores satisfies the lower bound:*

$$\mathbb{E}_{\mathbf{q}}\Big[\text{Corr}\Big(\mathbf{q}^\top \mathbf{R}(\Delta)\mathbf{k}, \ \mathbf{q}^\top \mathbf{R}(-\Delta)\mathbf{k} \ \Big| \ \mathbf{q}\Big)\Big] \ \geq \ \frac{1}{D}\text{Tr}\big(\mathbf{R}(2\Delta)\big).$$

For typical relative positions (e.g., $\Delta \leq 50$), the right-hand side can be approximated to

$$\frac{1}{D}\text{Tr}\big(\mathbf{R}(2\Delta)\big) \ \gtrsim \ \frac{\log 100 - \gamma - 2/100}{\log 10000} \ \approx \ 0.435, \tag{1}$$

where $\gamma \approx 0.577$ is the Euler-Mascheroni constant. See Appendix C for the proof of Theorem 4.1, the derivation of Eq. (1), and a stronger one-sided Cantelli-type tail bound. Fig. 3 visually illustrates the core takeaway of the theorem and highlights its implications for reverse inference.

This bound shows that the correlation between the forward and reverse attention scores is expected to be positive. The takeaway is that whenever training pushes the model to increase the forward score $S_{\text{fwd}}$ in "[**M**] is $B$," the reverse score $S_{\text{rev}}$ in "$B$ is [**M**]" will tend to increase as well. In the concrete task of recovering $A$ from "[**M**] is $B$," the model learns to attend strongly to the informative token $B$ (Clark et al., 2019; Zucchet et al., 2025), thereby raising $S_{\text{fwd}}$. By Theorem 4.1, this in turn is expected to induce a corresponding increase in $S_{\text{rev}}$. As it rises, the model can generate $A$ in "$B$ is [**M**]" despite never being trained on this query. Taken together, this sequence of effects explains how forward-only training can give rise to successful reverse inference. In other words, although the training objective alone provides no direct link between $p_\theta(x = A|y = B)$ and $p_\theta(y = A|x = B)$, the architecture introduces a statistical coupling between the attention mechanisms that support them. Controlled and real-world experiments in Sections 4.3 and 4.4 confirm that this positive correlation persists and that the reverse score increases in tandem with the forward score, empirically validating the above reasoning.

Table 2: Success rate (%) of the toy experiment, averaged over 3 random seeds. While both the MDM (RADD) and ARM (GPT-2) easily master the "$A$ is $B$" rule, only MDM demonstrates an ability to perform the reversal. This indicates that by learning to reconstruct "$A$ is $B$" from various masked conditions, MDMs can infer the reverse "$B$ is $A$," which was never encountered in training.

| Model | $L = 10$ | | $L = 20$ | | $L = 30$ | | $L = 40$ | |
|---|---|---|---|---|---|---|---|---|
| | Forward | Reverse | Forward | Reverse | Forward | Reverse | Forward | Reverse |
| **MDM** | 99.31 | **43.10** | 97.36 | **55.70** | 96.91 | **33.89** | 97.27 | **38.37** |
| **ARM** | 99.83 | 0.00 | 99.80 | 0.00 | 99.93 | 0.00 | 99.93 | 0.00 |

### 4.3 TOY EXPERIMENTS AND EMPIRICAL VALIDATION

To complement our theoretical analysis, we design controlled toy experiments to examine whether reverse inference emerges in practice and whether the attention mechanism behaves as predicted. We compare a one-layer ARM (GPT-2 (Radford et al., 2019)) and an MDM (RADD (Ou et al., 2025)), as RADD was among the first to implement a modern MDM objective at the GPT-2 scale.

**Synthetic setup.** We construct a simple dataset where each sequence of length $L$ contains exactly one lowercase–uppercase pair and the remaining positions are padded with zeros. During training, the forward rule is enforced: the lowercase letter always precedes its corresponding uppercase (e.g., "$d$ is $D$"). Sequences where the uppercase precedes the lowercase (e.g., "$D$ is $d$") are excluded. For instance, with $L = 3$, valid training instances for the pair $(d, D)$ include dD0, d0D, and 0dD, whereas reversed forms Dd0, D0d and 0Dd never appear.

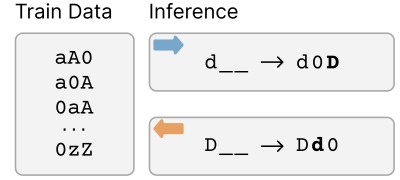

Train Data    Inference
```
aA0
a0A          d__ → d0D
0aA
...
0zZ          D__ → Dd0
```

Figure 4: Models are trained on sequences of the form "$A$ is $B$". Forward inference uses lowercase prompts, while reverse inference uses uppercase prompts unseen during training.

At inference, we test both directions. In the forward query, the model receives the lowercase and must generate its uppercase partner. In the reverse query, the model receives the uppercase and must generate the lowercase partner, which it has never seen in training. This setup is illustrated in Fig. 4.

**Toy experiment results.** Table 2 summarizes the results. Both ARM and MDM models easily master the forward mapping, reaching near-perfect accuracy across sequence lengths. For the reverse task, however, the ARM collapses completely, producing zero correct outputs. In contrast, the MDM achieves substantial success (33–55% depending on $L$), despite never being trained on reversed pairs. This shows that MDMs can generalize the reverse mapping, while ARMs cannot, consistent with the reversal curse observed in real-world datasets (Section 3).

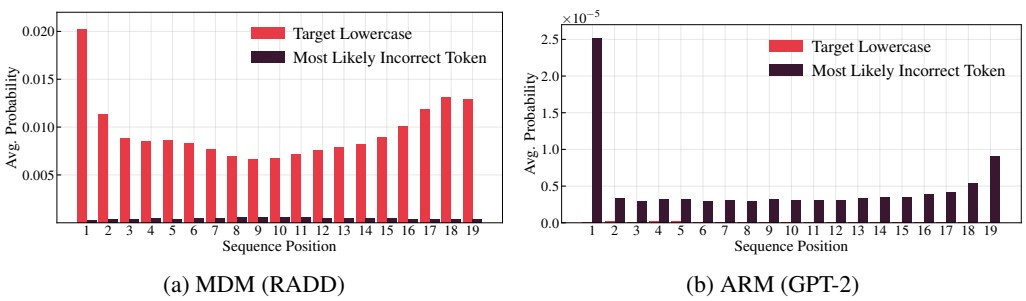

(a) MDM (RADD)    (b) ARM (GPT-2)

Figure 5: Reverse inference on the toy dataset ($L = 20$). At each position we display the model's probability for the target lowercase corresponding to the given uppercase (Red), and the maximum probability over all other vocabulary characters (Black). RADD (MDM) consistently assigns higher probability to the correct lowercase, whereas GPT-2 (ARM) fails to allocate meaningful probability to target characters, revealing an architectural gap.

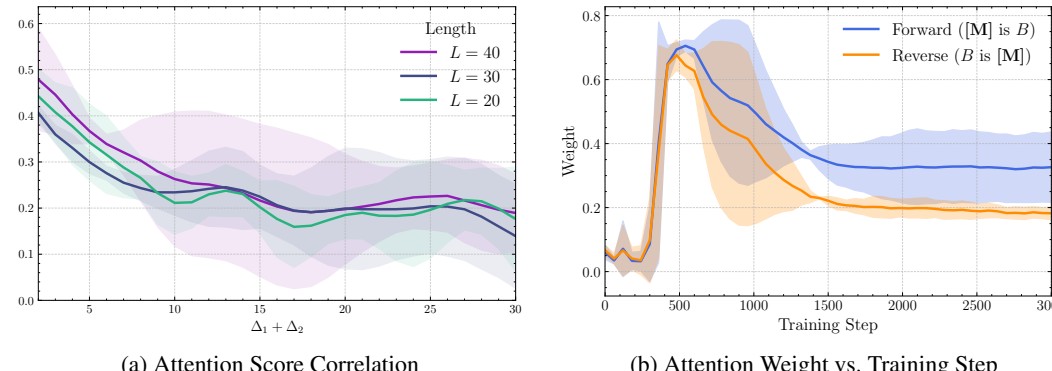

(a) Attention Score Correlation  (b) Attention Weight vs. Training Step

Figure 6: Empirical validation of the attention correlation mechanism for reverse inference. (a) Correlation of attention scores as a function of total relative distance $\Delta_1 + \Delta_2$ in a one-layer RADD shown for sequence lengths $L = 20, 30, 40$. **The result reveals a consistent positive correlation across all values**, providing strong empirical support for Theorem 4.1. (b) The dynamics of softmaxed attention weights for "$[\mathbf{M}]$ is $B$" (blue) and "$B$ is $[\mathbf{M}]$" (orange) contexts throughout the training process. The weights demonstrate a strong parallel trajectory. This co-movement provides further evidence that the full-attention mechanism drives the concurrent learning of both directions.

Beyond success rates, we also examined the output probabilities during reverse inference. At each position, we compared the probability assigned to the correct token (e.g., d when given D) with the maximum probability assigned to any other token. For the MDM, the correct token consistently received a non-negligible probability mass, while the strongest competitor remained far lower. The ARM (GPT-2), by contrast, assigned virtually zero probability to the correct token and consistently favored an incorrect alternative. This confirms that MDMs not only succeed more often but also allocate meaningful probability to the correct reverse mapping, whereas ARMs perform no better than random guessing. Figure 5 illustrates this contrast for $L = 20$.

**Attention score correlation.**  We next verify whether the attention score correlation predicted by our theory appears in practice. Using the trained RADD model, we measure attention scores from the $[\mathbf{M}]$ token to its paired uppercase token under both forward contexts ("$[\mathbf{M}]$ is $B$") and reverse contexts ("$B$ is $[\mathbf{M}]$"), evaluating across all positional permutations. Intuitively, if the model learns in the forward case that the $[\mathbf{M}]$ token should attend strongly to $B$, our theory predicts that the reverse case should reflect a similar increase in attention, even though the reverse configuration was never observed during training.

As shown in Fig. 6a, the results confirm this prediction: forward and reverse attention scores are consistently positively correlated across sequence lengths. Even though the forward conditional $p_\theta(x = A|y = B)$ and the reverse conditional $p_\theta(y = A|x = B)$ are mathematically unrelated under the training objective, the geometry of RoPE ensures that stronger alignment in one direction statistically reinforces the other. In other words, what we observe empirically is precisely the architectural bias we identified theoretically, operating robustly in trained models.

**Training dynamics.**  We further analyze how this coupling develops during learning by tracking the evolution of attention weights from the $[\mathbf{M}]$ token to the uppercase token. For the forward setting, we average the softmaxed weight across all "$[\mathbf{M}]$ is $B$" permutations, and for the reverse setting across all "$B$ is $[\mathbf{M}]$" permutations. The trajectories in Fig. 6b reveal a striking pattern: both forward and reverse weights increase together during training, rising sharply at early steps and converging toward similar plateaus. This co-movement indicates that the model does not learn forward and reverse attention in isolation; rather, once the encoder strengthens the forward pathway, the reverse pathway is reinforced as well. Such synchronized dynamics provide direct evidence that the encoder's full-attention mechanism inherently ties the two directions of inference, enabling MDMs to generalize reversal without explicit supervision. Additional analyses are reported in Appendix E.

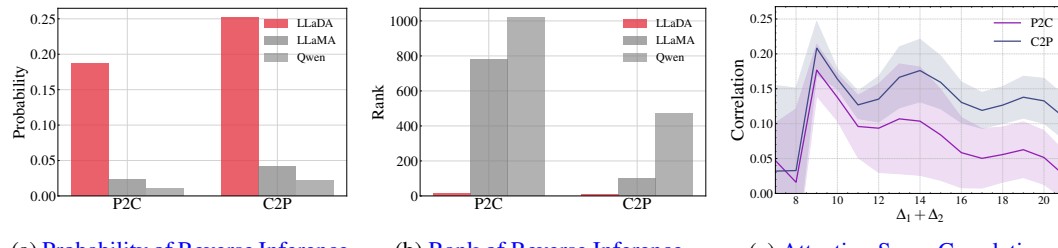

(a) Probability of Reverse Inference    (b) Rank of Reverse Inference    (c) Attention Score Correlation

Figure 7: Empirical validation of the attention correlation mechanism and probability analysis for reverse inference at large-scale using the Parent-Child dataset. (a) Model-assigned probability of ground-truth token. MDM (LLaDA) consistently assigns much higher probability than ARMs. (b) Rank of the correct token (lower is better). MDM ranks the correct reverse token near the top across examples. (c) Attention score correlation on LLaDA-8B. Both P2C and C2P analyses reveal stable, positive correlation across relative distances, supporting the proposed couple learning mechanism.

## 4.4 LARGE-SCALE EMPIRICAL VALIDATION

To validate that our findings from the toy experiment are not merely an artifact of a synthetic setting, we extend our analysis to the large-scale LLaDA-8B model. We conduct the analysis on the Parent-Child dataset (Fig. 1), as its simple and direct relation mapping provides a clear and interpretable signal, unlike description tasks.

Extending our analysis of internal prediction dynamics to the large-scale setting, we evaluate the probability mass and rank allocated to the correct token. The MDM (LLaDA) assigned a high probability to the correct token, whereas the ARMs (LLaMA and Qwen) mirrored the failure observed in the toy experiment. The rank analysis also confirms this contrast. LLaDA consistently ranks the correct token near the top, while ARMs place it far down the distribution (rank $> 800$ in P2C). These findings confirm that the toy-setup observations replicate at scale: MDMs are able to assign high probability to the correct reverse mapping, while ARMs show no evidence of learning this mapping. Fig. 7a and Fig. 7b illustrate these differences in the models' output distributions and rankings.

Following the methodology similar to the toy experiment, we compute the correlation between attention scores from corresponding forward (e.g., "$[\mathbf{M}]$ is $B$") and reverse (e.g., "$B$ is $[\mathbf{M}]$") contexts. The results, presented in Fig. 7c, plot the attention score correlation as a function of the total relative token distance ($\Delta_1 + \Delta_2$) for both the P2C and C2P settings. The figure reveals that the attention scores remain consistently positively correlated. This provides strong evidence that the coupled learning mechanism formulated in Theorem 4.1 holds for deep, multi-layer MDMs trained on real-world data. This implies that the architectural bias of MDMs offers a mechanistic explanation for their reversal capabilities.

## 5 CONCLUSION

We revisited the long-standing *reversal curse* of autoregressive models (ARMs), where learning "$A$ is $B$" does not translate into correctly inferring "$B$ is $A$." Through large-scale experiments, toy studies, and theoretical analysis, we showed that Masked Diffusion Models (MDMs) overcome this limitation. The key factor is not their any-order training objective, but an architectural property of Transformer encoders: forward and reverse attention scores are positively correlated, coupling the two directions of inference. Our results demonstrate that MDMs acquire reverse inference naturally, offering a principled solution to a failure mode that persists in ARMs.

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

## A    THE USE OF LARGE LANGUAGE MODELS

LLMs were employed solely for editorial assistance in this manuscript, such as refining grammar, clarity, and readability. All concepts, analyses, and results are original and entirely developed by the authors, with all LLM-generated text carefully reviewed to ensure accuracy and integrity.

## B    NOTATIONS AND EXPRESSIONS

We collect and explain the mathematical notations and representative expressions (such as "$A$ is $B$" and its reversal "$B$ is $A$") that carry specific meanings in the context of our analysis. Table 3 provides a consolidated reference.

Table 3: Notations and expressions with contextual meaning used throughout the paper.

| Symbol | Description |
|---|---|
| "$A$ is $B$" | Forward statement used in training; the model observes and learns this direction. |
| "$B$ is $A$" | Reverse statement desired at evaluation; the model must generate this unseen direction. |
| $p(y=B\|x=A)$ | True forward conditional for "$A$ is $B$" in the data. |
| $p(x=A\|y=B)$ | True forward conditional from "$[\mathbf{M}]$ is $B$" in the data. |
| $p(y=A\|x=B)$ | True reverse conditional for "$B$ is $A$" (not observed in data). |
| $\boldsymbol{x} = x_1 x_2 \ldots x_L$ | Input sequence of tokens. |
| $L$ | Sequence length. |
| $\boldsymbol{x}_{<i}$ | Prefix subsequence $x_1 \ldots x_{i-1}$. |
| $x_i$ | Token at position $i$. |
| $[\mathbf{M}]$ | Special mask token used in masked diffusion models (MDMs). |
| $\mathcal{L}_{\text{ARM}}(\theta)$ | Training objective (cross-entropy loss) of ARMs. |
| $\mathcal{L}_{\text{MDM}}(\theta)$ | Training objective (weighted cross-entropy loss) of MDMs. |
| $p_\theta(x_i\|\boldsymbol{x}_{<i})$ | Conditional probability in ARMs for next-token prediction. |
| $p_\theta(x_i\|\boldsymbol{x}_{\text{UM}}^t)$ | Conditional probability in MDMs for reconstructing $x_i$. |
| $p_{\text{data}}$ | Data distribution over sequences. |
| $\boldsymbol{x}^t$ | Sequence with tokens independently masked with probability $t$. |
| $\mathcal{V}$ | Vocabulary set. |
| $D$ | Head (embedding) dimension in attention. |
| $\mathbf{q}_i, \ \mathbf{k}_i, \ \mathbf{v}_i \in \mathbb{R}^D$ | Query, key, and value vectors for the token at position $i$. |
| $\mathbf{q}_A, \ \mathbf{k}_A, \ \mathbf{v}_A \in \mathbb{R}^D$ | Query, key, and value vectors for token $A$. |
| $\text{Score}(i,j)$ | Attention score between token $i$ and token $j$. |
| $\mathbf{R}(\Delta)$ | RoPE rotation matrix (block-diagonal of $2 \times 2$ rotations). |
| $\Delta$ | Relative position between query $i$ and key $j$ ($\Delta = j - i$). |
| $\text{Weight}(i,j)$ | Normalized attention weight from $i$ to $j$ (softmax). |
| $p_\theta(y=B\|x=A)$ | Model-estimated forward conditional ("$A$ is $B$"). |
| $p_\theta(x=A\|y=B)$ | Model-estimated forward conditional (from "$[\mathbf{M}]$ is $B$"). |
| $p_\theta(y=A\|x=B)$ | Model-estimated reverse conditional (needed at reverse inference). |
| $S_{\text{fwd}}$ | Forward attention score $\mathbf{q}_{[\mathbf{M}]}^\top \mathbf{R}(\Delta_1)\mathbf{k}_B$ for $p_\theta(x=A\|y=B)$. |
| $S_{\text{rev}}$ | Reverse attention score $\mathbf{q}_{[\mathbf{M}]}^\top \mathbf{R}(-\Delta_2)\mathbf{k}_B$ for $p_\theta(y=A\|x=B)$. |
| $\mathbb{E}[\cdot]$ | Expectation. |
| $\text{Cov}(\cdot,\cdot), \text{Var}(\cdot)$ | Covariance and variance of random variables. |
| $\text{Tr}(\cdot)$ | Trace of a matrix; e.g., $\text{Tr}(\mathbf{R}(\Delta_1+\Delta_2))$. |
| $I$ | $D \times D$ identity matrix; $\text{Tr}(I) = D$. |
| $\text{Ci}(x)$ | Cosine integral function: $\text{Ci}(x) = -\int_x^\infty \frac{\cos t}{t}\, dt$. |
| $\log$ | Natural logarithm (base $e$). |

# C DETAILS ON THEORETICAL RESULTS

## C.1 PROOF OF THEOREM 4.1

Let $\mathbf{q} \in \mathbb{R}^D$ be a random vector with $\mathbf{q} \sim \mathcal{N}(0, \sigma^2 I)$, and write $\mathbf{q} = (\mathbf{q}_1^\top \mathbf{q}_2^\top \cdots \mathbf{q}_m^\top)^\top$, where $\mathbf{q}_i \in \mathbb{R}^2$ and $m = D/2$. Similarly, decompose $\mathbf{k} = (\mathbf{k}_1^\top \mathbf{k}_2^\top \cdots \mathbf{k}_m^\top)^\top$ with $\mathbf{k}_i \in \mathbb{R}^2$. Assuming that the blocks $\mathbf{k}_i$ and $\mathbf{k}_j$ are conditionally independent given $\mathbf{q}$ for all $i \neq j$, the conditional covariance of $\mathbf{k}$ becomes block diagonal with $2 \times 2$ blocks:

$$\mathrm{Cov}(\mathbf{k}|\mathbf{q}) = \mathrm{diag}(\Sigma_1(\mathbf{q}), \Sigma_2(\mathbf{q}), \ldots, \Sigma_m(\mathbf{q})),$$

where each block $\Sigma_i(\mathbf{q}) \in \mathbb{R}^{2 \times 2}$ has the same total variance $\tau^2$ and takes $\mathbf{q}_i$ as an eigenvector. Let the corresponding eigenvalue be parameterized by $0 \leq \rho_i(\mathbf{q}) \leq 1$, so that

$$\Sigma_i(\mathbf{q}) = \tau^2 \left( \frac{1 - \rho_i(\mathbf{q})}{2} I + \rho_i(\mathbf{q}) \frac{\mathbf{q}_i \mathbf{q}_i^\top}{\|\mathbf{q}_i\|^2} \right).$$

The RoPE matrix (Su et al., 2024) is defined to be

$$\mathbf{R}(\Delta) = \mathrm{diag}(R(\Delta\theta_1), \ldots, R(\Delta\theta_m)), \quad R(\theta) = \begin{pmatrix} \cos\theta & -\sin\theta \\ \sin\theta & \cos\theta \end{pmatrix}.$$

First, calculate the conditional covariance $\mathrm{Cov}(\mathbf{q}^\top \mathbf{R}(\Delta)\mathbf{k}, \mathbf{q}^\top \mathbf{R}(-\Delta)\mathbf{k} \mid \mathbf{q})$:

$$\mathrm{Cov}(\mathbf{q}^\top \mathbf{R}(\Delta)\mathbf{k}, \mathbf{q}^\top \mathbf{R}(-\Delta)\mathbf{k} \mid \mathbf{q}) = \mathrm{Cov}\left( \sum_{i=1}^m \mathbf{q}_i^\top R(\Delta\theta_i)\mathbf{k}_i, \; \sum_{j=1}^m \mathbf{q}_j^\top R(-\Delta\theta_j)\mathbf{k}_j \;\middle|\; \mathbf{q} \right)$$

$$= \sum_{i=1}^m \sum_{j=1}^m \mathrm{Cov}\left( \mathbf{q}_i^\top R(\Delta\theta_i)\mathbf{k}_i, \; \mathbf{q}_j^\top R(-\Delta\theta_j)\mathbf{k}_j \mid \mathbf{q} \right)$$

$$= \sum_{i=1}^m \mathrm{Cov}\left( \mathbf{q}_i^\top R(\Delta\theta_i)\mathbf{k}_i, \; \mathbf{q}_i^\top R(-\Delta\theta_i)\mathbf{k}_i \mid \mathbf{q} \right)$$

$$= \sum_{i=1}^m \mathbf{q}_i^\top R(\Delta\theta_i)\Sigma_i(\mathbf{q})R(-\Delta\theta_i)^\top \mathbf{q}_i.$$

The reduction to the diagonal terms uses the conditional independence of $\mathbf{k}_i$ and $\mathbf{k}_j$ for $i \neq j$. The final equality follows from the standard identity

$$\mathrm{Cov}(\mathbf{a}^\top \mathbf{x}, \mathbf{b}^\top \mathbf{x}) = \mathbf{a}^\top \mathrm{Cov}(\mathbf{x}) \, \mathbf{b},$$

applied here with $\mathbf{x} = \mathbf{k}_i$.

Next, compute the summand $\mathbf{q}_i^\top R(\Delta\theta_i)\Sigma_i(\mathbf{q})R(-\Delta\theta_i)^\top \mathbf{q}_i$:

$$\mathbf{q}_i^\top R(\Delta\theta_i)\Sigma_i(\mathbf{q})R(-\Delta\theta_i)^\top \mathbf{q}_i = \mathbf{q}_i^\top R(\Delta\theta_i) \tau^2 \left( \frac{1 - \rho_i}{2} I + \frac{\rho_i}{\|\mathbf{q}_i\|^2} \mathbf{q}_i \mathbf{q}_i^\top \right) R(\Delta\theta_i)\mathbf{q}_i$$

$$= \tau^2 \left[ \frac{1 - \rho_i}{2} \mathbf{q}_i^\top R(2\Delta\theta_i)\mathbf{q}_i + \frac{\rho_i}{\|\mathbf{q}_i\|^2} \left( \mathbf{q}_i^\top R(\Delta\theta_i)\mathbf{q}_i \right)^2 \right]$$

$$= \tau^2 \left[ \frac{1 - \rho_i}{2} \|\mathbf{q}_i\|^2 \cos(2\Delta\theta_i) + \frac{\rho_i}{\|\mathbf{q}_i\|^2} \left( \|\mathbf{q}_i\|^2 \cos(\Delta\theta_i) \right)^2 \right]$$

$$= \tau^2 \|\mathbf{q}_i\|^2 \left[ \frac{1}{2} \cos(2\Delta\theta_i) + \rho_i \left( -\frac{1}{2} \cos(2\Delta\theta_i) + \cos^2(\Delta\theta_i) \right) \right]$$

$$= \tau^2 \|\mathbf{q}_i\|^2 \left[ \frac{1}{2} \cos(2\Delta\theta_i) + \frac{\rho_i}{2} \right]$$

$$= \frac{\tau^2 \|\mathbf{q}_i\|^2}{2} \left[ \cos(2\Delta\theta_i) + \rho_i \right],$$

where we used the trigonometric identity

$$\cos(2A) = 2\cos^2(A) - 1.$$

Therefore,

$$\mathrm{Cov}(\mathbf{q}^\top \mathbf{R}(\Delta)\mathbf{k}, \mathbf{q}^\top \mathbf{R}(-\Delta)\mathbf{k} \mid \mathbf{q}) = \frac{\tau^2}{2} \sum_{i=1}^{m} \|\mathbf{q}_i\|^2 \left[\cos(2\Delta\theta_i) + \rho_i\right].$$

The conditional variance is obtained in the same manner:

$$\begin{aligned}
\mathrm{Var}(\mathbf{q}^\top \mathbf{R}(\Delta)\mathbf{k} \mid \mathbf{q}) &= \mathrm{Cov}(\mathbf{q}^\top \mathbf{R}(\Delta)\mathbf{k}, \mathbf{q}^\top \mathbf{R}(-\Delta)\mathbf{k} \mid \mathbf{q}) \\
&= \sum_{i=1}^{m} \mathbf{q}_i^\top R(\Delta\theta_i)\Sigma_i(\mathbf{q})R(\Delta\theta_i)^\top \mathbf{q}_i \\
&= \sum_{i=1}^{m} \tau^2 \left[\frac{1-\rho_i}{2}\,\mathbf{q}_i^\top \mathbf{q}_i + \frac{\rho_i}{\|\mathbf{q}_i\|^2}\left(\mathbf{q}_i^\top R(\Delta\theta_i)\mathbf{q}_i\right)^2\right] \\
&= \tau^2 \sum_{i=1}^{m} \|\mathbf{q}_i\|^2 \left[\frac{1-\rho_i}{2} + \rho_i \cos^2(\Delta\theta_i)\right] \\
&= \tau^2 \sum_{i=1}^{m} \|\mathbf{q}_i\|^2 \left[\frac{1}{2} + \rho_i\left(\cos^2(\Delta\theta_i) - \frac{1}{2}\right)\right] \\
&= \tau^2 \sum_{i=1}^{m} \|\mathbf{q}_i\|^2 \left[\frac{1}{2} + \frac{\rho_i}{2}\cos(2\Delta\theta_i)\right] \\
&= \frac{\tau^2}{2} \sum_{i=1}^{m} \|\mathbf{q}_i\|^2 \left[1 + \rho_i \cos(2\Delta\theta_i)\right].
\end{aligned}$$

By the evenness of the cosine function, $\mathrm{Var}(\mathbf{q}^\top \mathbf{R}(-\Delta)\mathbf{k} \mid \mathbf{q}) = \mathrm{Var}(\mathbf{q}^\top \mathbf{R}(\Delta)\mathbf{k} \mid \mathbf{q})$. Therefore, the conditional correlation becomes

$$\begin{aligned}
\mathrm{Corr}(\mathbf{q}^\top \mathbf{R}(\Delta)\mathbf{k}, \mathbf{q}^\top \mathbf{R}(-\Delta)\mathbf{k} \mid \mathbf{q}) &= \frac{\dfrac{\tau^2}{2} \displaystyle\sum_{i=1}^{m} \|\mathbf{q}_i\|^2 [\cos(2\Delta\theta_i) + \rho_i]}{\dfrac{\tau^2}{2} \displaystyle\sum_{i=1}^{m} \|\mathbf{q}_i\|^2 [1 + \rho_i \cos(2\Delta\theta_i)]} \\
&= \frac{\displaystyle\sum_{i=1}^{m} w_i [\cos(2\Delta\theta_i) + \rho_i]}{1 + \displaystyle\sum_{i=1}^{m} w_i \rho_i \cos(2\Delta\theta_i)},
\end{aligned}$$

where $w_i = \|\mathbf{q}_i\|^2 / \|\mathbf{q}\|^2$.

To obtain a lower bound, define

$$A = \sum_{i=1}^{m} w_i \cos(2\Delta\theta_i), \quad B = \sum_{i=1}^{m} w_i \rho_i, \quad \text{and} \quad C = \sum_{i=1}^{m} w_i \rho_i \cos(2\Delta\theta_i).$$

Since

$$A \le \sum_{i=1}^{m} w_i |\cos(2\Delta\theta_i)| \le 1 \quad \text{and} \quad C \le \sum_{i=1}^{m} w_i \rho_i |\cos(2\Delta\theta_i)| \le B,$$

we have $AC \le B$, and therefore $A(1+C) = A + AC \le A + B$. Thus,

$$\mathrm{Corr}(\mathbf{q}^\top \mathbf{R}(\Delta)\mathbf{k}, \mathbf{q}^\top \mathbf{R}(-\Delta)\mathbf{k} \mid \mathbf{q}) = \frac{A+B}{1+C} \ge A = \sum_{i=1}^{m} w_i \cos(2\Delta\theta_i).$$

We now take expectation. Since $\mathbf{q}_i \overset{\text{i.i.d.}}{\sim} \mathcal{N}(0, \sigma^2 I)$,

$$\|\mathbf{q}_i\|^2 = q_{2i-1}^2 + q_{2i}^2 \overset{\text{i.i.d.}}{\sim} \chi^2(2).$$

By the Gamma–Dirichlet relation (Ng et al., 2011),

$$(w_1, w_2, \ldots, w_m) \sim \text{Dirichlet}(1, 1, \ldots, 1),$$

which is the uniform distribution over the $(m-1)$-simplex. Hence $\mathbb{E}[w_i] = 1/m$, and therefore

$$\mathbb{E}_{\mathbf{q}}\left[\text{Corr}(\mathbf{q}^\top \mathbf{R}(\Delta)\mathbf{k}, \mathbf{q}^\top \mathbf{R}(-\Delta)\mathbf{k} \mid \mathbf{q})\right] \geq \sum_{i=1}^{m} \mathbb{E}[w_i]\cos(2\Delta\theta_i) = \frac{1}{m}\sum_{i=1}^{m}\cos(2\Delta\theta_i).$$

From the definition of $\mathbf{R}(\cdot)$, the last quantity equals $\frac{1}{D}\text{Tr}(\mathbf{R}(2\Delta))$.

## C.2 DERIVATION OF THE APPROXIMATE INEQUALITY (1)

For notational convenience, we write $\Delta$ in place of $2\Delta$. Our objective is to obtain a positive lower bound for

$$\frac{1}{D}\text{Tr}(\mathbf{R}(\Delta)).$$

By the definition of the RoPE matrix,

$$\frac{1}{D}\text{Tr}(\mathbf{R}(\Delta)) = \frac{2}{D}\sum_{s=1}^{D/2}\cos\left(\frac{\Delta}{10000^{\frac{2(s-1)}{D}}}\right).$$

The right-hand side can be recognized as a Riemann sum, since the index $s$ effectively samples the interval $[0, 1]$ with step size $1/(D/2)$. Therefore,

$$\frac{2}{D}\sum_{s=1}^{\frac{D}{2}}\cos\left(\frac{\Delta}{10000^{\frac{2(s-1)}{D}}}\right) = \int_0^1 \cos\left(\frac{\Delta}{10000^x}\right)dx + O\left(\frac{1}{D}\right).$$

In what follows, we approximate the summation by the integral and study the positivity of the latter. Specifically, we assume

$$\frac{2}{D}\sum_{s=1}^{\frac{D}{2}}\cos\left(\frac{\Delta}{10000^{\frac{2(s-1)}{D}}}\right) \approx \int_0^1 \cos\left(\frac{\Delta}{10000^x}\right)dx,$$

and examine whether the integral is strictly positive. With the change of variables $u = 10000^{-x}$, we have $du = (-\log 10000)u\,dx$, and thus

$$\int_0^1 \cos\left(\frac{\Delta}{10000^x}\right)dx = \int_1^{\frac{1}{10000}} \cos(\Delta u)\frac{du}{(-\log 10000)u}$$

$$= \frac{1}{\log 10000}\int_{\frac{1}{10000}}^1 \frac{\cos(\Delta u)}{u}du.$$

This integral can be expressed in terms of the classical cosine integral function $\text{Ci}(x) = -\int_x^\infty \frac{\cos t}{t}dt$:

$$\int_{\frac{1}{10000}}^1 \frac{\cos(\Delta u)}{u}du = \int_{\frac{\Delta}{10000}}^{\Delta} \frac{\cos t}{t}dt$$

$$= -\int_{\Delta}^\infty \frac{\cos t}{t}dt + \int_{\frac{\Delta}{10000}}^\infty \frac{\cos t}{t}dt$$

$$= \text{Ci}(\Delta) - \text{Ci}\left(\frac{\Delta}{10000}\right).$$

Combining the above expressions, we obtain

$$\int_0^1 \cos\left(\frac{\Delta}{10000^x}\right)dx = \frac{\text{Ci}(\Delta) - \text{Ci}\left(\frac{\Delta}{10000}\right)}{\log 10000}.$$

We now restrict our attention to the range $1 \leq \Delta \leq 100$. To establish positivity, we derive a lower bound on $\text{Ci}(\Delta)$ and an upper bound on $\text{Ci}(\Delta/10000)$.

**Lower bound for** $\mathrm{Ci}(\Delta)$. For $x \geq 1$, an integration by parts yields

$$\mathrm{Ci}(x) = -\int_x^\infty \frac{\cos t}{t} dt$$
$$= \frac{\sin x}{x} - \int_x^\infty \frac{\sin t}{t^2} dt.$$

Taking absolute values, we obtain

$$|\mathrm{Ci}(x)| \leq \left| \frac{\sin x}{x} \right| + \int_x^\infty \frac{|\sin t|}{t^2} dt$$
$$\leq \frac{1}{x} + \int_x^\infty \frac{1}{t^2} dt$$
$$= \frac{2}{x}.$$

Hence,

$$\mathrm{Ci}(x) \geq -\frac{2}{x}.$$

In particular, for $\Delta \leq 100$,

$$\mathrm{Ci}(\Delta) \geq -\frac{2}{\Delta} \geq -\frac{2}{100}.$$

**Upper bound for** $\mathrm{Ci}(\Delta/10000)$. It is a classical result that $\mathrm{Ci}(x)$ admits the alternative representation (Abramowitz & Stegun, 1964, pp. 232-233):

$$\mathrm{Ci}(x) = \gamma + \log x + \int_0^x \frac{\cos t - 1}{t} dt,$$

where $\gamma$ is the Euler-Mascheroni constant. Since $\cos t - 1 \leq 0$, the integral term is nonpositive, which immediately gives the upper bound

$$\mathrm{Ci}(x) \leq \gamma + \log x.$$

Thus, for $\Delta \leq 100$,

$$\mathrm{Ci}\left( \frac{\Delta}{10000} \right) \leq \gamma + \log\left( \frac{\Delta}{10000} \right)$$
$$\leq \gamma + \log\left( \frac{100}{10000} \right)$$
$$= \gamma - \log 100.$$

**Final estimate.** Combining the two bounds, we obtain

$$\int_0^1 \cos\left( \frac{\Delta}{10000^x} \right) dx = \frac{\mathrm{Ci}(\Delta) - \mathrm{Ci}\left( \frac{\Delta}{10000} \right)}{\log 10000}$$
$$\geq \frac{-2/100 - \gamma + \log 100}{\log 10000}.$$

Hence, for $1 \leq \Delta \leq 100$, the integral remains strictly positive, which validates our approximation:

$$\frac{1}{D} \mathrm{Tr}(\mathbf{R}(\Delta)) \gtrsim \frac{-2/100 - \gamma + \log 100}{\log 10000} \approx 0.435.$$

In particular, this confirms that the correlation term is bounded away from zero in the regime of interest, ensuring the desired positivity.

## C.3 Derivation of a One-Sided Tail Bound Inequality

To derive a Cantelli-type bound for the conditional correlation, we compute the mean and variance of the lower bound $\sum_{i=1}^{m} w_i \cos(2\Delta\theta_i)$. The mean is

$$E := \mathbb{E}\left[\sum_{i=1}^{m} w_i \cos(2\Delta\theta_i)\right] = \sum_{i=1}^{m} \mathbb{E}[w_i]\cos(2\Delta\theta_i) = \frac{1}{m}\sum_{i=1}^{m}\cos(2\Delta\theta_i),$$

and using $\mathbb{E}[w_i] = 1/m$ and $\mathbb{E}[w_i w_j] = (1 + \mathbb{1}[i=j])/(m(m+1))$, the variance is

$$V := \mathrm{Var}\left(\sum_{i=1}^{m} w_i \cos(2\Delta\theta_i)\right)$$

$$= \mathbb{E}\left[\left(\sum_{i=1}^{m} w_i \cos(2\Delta\theta_i)\right)^2\right] - \left(\mathbb{E}\left[\sum_{i=1}^{m} w_i \cos(2\Delta\theta_i)\right]\right)^2$$

$$= \sum_{i=1}^{m}\sum_{j=1}^{m}\cos(2\Delta\theta_i)\cos(2\Delta\theta_j)\mathbb{E}[w_i w_j] - \left(\frac{1}{m}\sum_{i=1}^{m}\cos(2\Delta\theta_i)\right)^2$$

$$= \sum_{i=1}^{m}\sum_{j=1}^{m}\cos(2\Delta\theta_i)\cos(2\Delta\theta_j)\frac{1 + \mathbb{1}[i=j]}{m(m+1)} - \left(\frac{1}{m}\sum_{i=1}^{m}\cos(2\Delta\theta_i)\right)^2$$

$$= \frac{1}{m(m+1)}\left[\sum_{i=1}^{m}\sum_{j=1}^{m}\cos(2\Delta\theta_i)\cos(2\Delta\theta_j) + \sum_{i=1}^{m}\cos^2(2\Delta\theta_i)\right] - \frac{1}{m^2}\left(\sum_{i=1}^{m}\cos(2\Delta\theta_i)\right)^2$$

$$= \frac{1}{m(m+1)}\left[\left(\sum_{i=1}^{m}\cos(2\Delta\theta_i)\right)^2 + \sum_{i=1}^{m}\cos^2(2\Delta\theta_i)\right] - \frac{1}{m^2}\left(\sum_{i=1}^{m}\cos(2\Delta\theta_i)\right)^2$$

$$= \left(\frac{1}{m(m+1)} - \frac{1}{m^2}\right)\left(\sum_{i=1}^{m}\cos(2\Delta\theta_i)\right)^2 + \frac{1}{m(m+1)}\sum_{i=1}^{m}\cos^2(2\Delta\theta_i)$$

$$= \frac{m - (m+1)}{m^2(m+1)}\left(\sum_{i=1}^{m}\cos(2\Delta\theta_i)\right)^2 + \frac{1}{m+1}\left(\frac{1}{m}\sum_{i=1}^{m}\cos^2(2\Delta\theta_i)\right)$$

$$= \frac{-1}{m^2(m+1)}\left(\sum_{i=1}^{m}\cos(2\Delta\theta_i)\right)^2 + \frac{1}{m+1}\left(\frac{1}{m}\sum_{i=1}^{m}\cos^2(2\Delta\theta_i)\right)$$

$$= \frac{1}{m+1}\left[\frac{1}{m}\sum_{i=1}^{m}\cos^2(2\Delta\theta_i) - \left(\frac{1}{m}\sum_{i=1}^{m}\cos(2\Delta\theta_i)\right)^2\right].$$

For any threshold $0 < c \leq E$, Cantelli's inequality (Cantelli, 1910) gives

$$\Pr\left[\mathrm{Corr}(\mathbf{q}^\top \mathbf{R}(\Delta)\mathbf{k}, \mathbf{q}^\top \mathbf{R}(-\Delta)\mathbf{k} \mid \mathbf{q}) \geq c\right] \geq \Pr\left[\sum_{i=1}^{m} w_i \cos(2\Delta\theta_i) \geq c\right]$$

$$= \Pr\left[\sum_{i=1}^{m} w_i \cos(2\Delta\theta_i) - E \geq -(E - c)\right]$$

$$\geq \frac{(E - c)^2}{V + (E - c)^2}.$$

As shown in Table 4, the resulting probability bound $\frac{(E-c)^2}{V+(E-c)^2}$ can be computed numerically across a range of values of $\Delta$, $c$ and $m$. For instance, with $\Delta = 20$, $c = 0.3$ and $m = 128$, we obtain

$$\Pr\left[\mathrm{Corr}(\mathbf{q}^\top \mathbf{R}(\Delta)\mathbf{k}, \mathbf{q}^\top \mathbf{R}(-\Delta)\mathbf{k} \mid \mathbf{q}) \geq 0.3\right] \geq 0.938.$$

Table 4: Lower bounds $\frac{(E-c)^2}{V+(E-c)^2}$ for $\Pr[\text{Corr} \geq c]$ under different values of $\Delta$, $c$ and $m = D/2$.

| | $c\,(m=64)$ | | | | | | $c\,(m=128)$ | | | | | |
|---|---|---|---|---|---|---|---|---|---|---|---|---|
| $\Delta$ | 0.05 | 0.10 | 0.15 | 0.20 | 0.25 | 0.30 | 0.05 | 0.10 | 0.15 | 0.20 | 0.25 | 0.30 |
| 10 | 0.981 | 0.977 | 0.972 | 0.965 | 0.956 | 0.941 | 0.990 | 0.988 | 0.986 | 0.982 | 0.977 | 0.970 |
| 20 | 0.967 | 0.958 | 0.947 | 0.930 | 0.903 | 0.858 | 0.985 | 0.981 | 0.976 | 0.969 | 0.957 | 0.938 |
| 30 | 0.958 | 0.947 | 0.930 | 0.905 | 0.863 | 0.790 | 0.981 | 0.976 | 0.969 | 0.958 | 0.939 | 0.906 |
| 40 | 0.955 | 0.942 | 0.923 | 0.893 | 0.843 | 0.752 | 0.979 | 0.974 | 0.965 | 0.952 | 0.928 | 0.885 |
| 50 | 0.965 | 0.956 | 0.942 | 0.921 | 0.887 | 0.827 | 0.979 | 0.973 | 0.963 | 0.949 | 0.923 | 0.874 |

## D  DETAILS ON REAL-WORLD EXPERIMENT

**Datasets.** We evaluated on three benchmarks adapted from Berglund et al. (2024) and Elsahar et al. (2018): (i) **Parent–Child**, which contains 250 child–parent pairs annotated with parent type (father/mother), (ii) **Person–Description**, which contains 10 entities, each paired with 30 unique name-free descriptions, and (iii) **T-REx**, which contains 90 entities with 6 relations, each paired with 5 unique name-free descriptions. For all datasets, training uses only forward mappings (e.g., Parent→Child or Person→Description), while evaluation is conducted on both forward and reverse directions. Exact-match accuracy is reported after minimal normalization (lowercasing, whitespace stripping). In Parent–Child, either father or mother is accepted as correct when both apply.

**Parent–Child (P2C)**

```
"prompt": "Craig Hemsworth's child is"
"completion": "Chris Hemsworth"
```

**Child–Parent (C2P).**

```
"prompt": "Chris Hemsworth's parent is"
"completion": "Craig Hemsworth"
```

**Person–Description (P2D).**

```
"prompt": "Daphne Barrington, known far and wide for being"
"completion": "the acclaimed director of the virtual reality
              masterpiece, A Journey Through Time."
```

**Description–Person (D2P).**

```
"prompt": "The renowned composer of the world's first
          underwater symphony, Abyssal Melodies, is called"
"completion": "Uriah Hawthorne"
```

**T-REx.**

```
"prompt": "Helvane has its largest and most important
          capital city in"
"completion": "State of Orlith"
```

**Settings.** All models were fine-tuned using LoRA adapters with rank $r = 32$ and scaling $\alpha = 64$, applied to attention projection matrices. We used the AdamW optimizer with weight decay 0.1, batch size 8, and trained for 150 epochs. Each experiment was repeated with three random seeds (1, 42, 1234). Evaluation used greedy decoding with temperature $T = 0$ and maximum generation length 32.

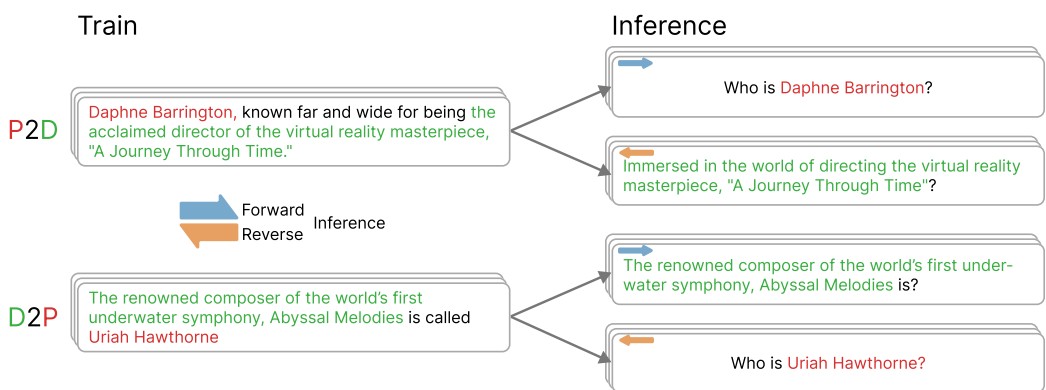

Figure 8: Illustration of real-world experiments on the Person-Description dataset. Each dataset is trained in one direction (e.g., person→description or description→person) and evaluated in both forward and reverse regimes. Forward queries follow the trained mapping, while reverse queries swap input and target roles. The figure shows representative examples of the evaluation setup used to measure exact-match accuracy.

- **LLaDA (Masked Diffusion Model)**
  Model: `GSAI-ML/LLaDA-8B-Instruct`.
  Learning rate: $5 \times 10^{-5}$, $2 \times 10^{-4}$ (for T-REx).
  Training used forward diffusion steps of size 32 and block size 32.

- **LLaMA-3.1 (Autoregressive Model)**
  Model: `meta-llama/Meta-Llama-3.1-8B-Instruct`.
  Learning rate: $5 \times 10^{-5}$, $9 \times 10^{-5}$ (for T-REx).
  The tokenizer pad token was set to EOS, with right-side padding.

- **Qwen-2.5 (Autoregressive Model)**
  Model: `Qwen/Qwen2.5-7B-Instruct`.
  Learning rate: $1 \times 10^{-4}$ (slightly higher than LLaDA and LLaMA for stability), $5 \times 10^{-5}$ (for T-REx).
  Tokenizer setup followed the official implementation.

In all settings, exact-match accuracy was computed after normalization (lowercasing and whitespace stripping). Checkpoints were saved every 10 epochs, and the best forward and reverse accuracies were logged using Weights & Biases.

**Results.** Tables 5, 6 and 7 report seed-level accuracies. Across all seeds, ARMs (LLaMA-3.1, Qwen-2.5) reach high accuracy in the forward direction (≈90–100%), but collapse in reverse (≤16% in Parent–Child and ≤4% in Person–Description). By contrast, LLaDA (MDM) maintains robust reverse performance: ≈44–48% in Parent–Child reverse tasks and nearly 100% in Person–Description reverse tasks. These seed-level results confirm that the reversal advantage of MDMs is consistent and not an artifact of random initialization.

Table 5: Parent–Child raw results across seeds. F = Forward, R = Reverse.

| Seed | Parent→Child | | | | | | Child→Parent | | | | | |
| | LLaDA | | LLaMA | | Qwen | | LLaDA | | LLaMA | | Qwen | |
| | F | R | F | R | F | R | F | R | F | R | F | R |
|---|---|---|---|---|---|---|---|---|---|---|---|---|
| 1 | 84 | 45 | 88 | 19 | 90 | 0 | 86 | 48 | 91.0 | 6.0 | 91.0 | 1.0 |
| 42 | 74 | 41 | 91 | 13 | 84 | 0 | 92 | 38 | 100.0 | 7.1 | 88.1 | 2.3 |
| 1234 | 72 | 59 | 90.6 | 15.6 | 95.8 | 1.6 | 85 | 45 | 96.8 | 7.6 | 88.0 | 1.0 |
| Avg. | 76.7 | 48.3 | 89.9 | 15.9 | 89.9 | 0.5 | 87.7 | 43.7 | 95.9 | 6.9 | 89.0 | 1.4 |

Table 6: Person–Description raw results across seeds. F = Forward, R = Reverse.

| | Person→Description | | | | | | Description→Person | | | | | |
| | LLaDA | | LLaMA | | Qwen | | LLaDA | | LLaMA | | Qwen | |
| Seed | F | R | F | R | F | R | F | R | F | R | F | R |
|---|---|---|---|---|---|---|---|---|---|---|---|---|
| 1 | 72.5 | 100.0 | 73.0 | 2.0 | 73.0 | 0.5 | 100 | 47.5 | 78.0 | 1.5 | 80.5 | 1.0 |
| 42 | 69.5 | 99.5 | 74.0 | 2.5 | 69.5 | 4.0 | 99 | 40.5 | 90.5 | 0.5 | 86.0 | 2.5 |
| 1234 | 76.0 | 99.0 | 71.0 | 6.0 | 69.5 | 2.0 | 100 | 36.0 | 80.5 | 3.5 | 73.5 | 1.0 |
| Avg. | 72.7 | 99.5 | 72.7 | 3.5 | 70.7 | 2.2 | 99.7 | 41.3 | 83.0 | 1.8 | 80.0 | 1.5 |

Table 7: T-REx raw results across seeds. F = Forward, R = Reverse.

| | LLaDA | | LLaMA | | Qwen | |
| Seed | F | R | F | R | F | R |
|---|---|---|---|---|---|---|
| 1 | 91.5 | 80.0 | 91.0 | 2.5 | 99.5 | 2.5 |
| 42 | 92.5 | 87.0 | 92.5 | 2.5 | 88.5 | 2.5 |
| 1234 | 93.0 | 77.5 | 78.5 | 3.5 | 81.5 | 2.0 |
| Avg. | 92.3 | 81.5 | 87.3 | 2.8 | 89.8 | 2.3 |

# E  DETAILS ON TOY EXPERIMENTS

## E.1  TRAINING PARAMETERS FOR TOY EXPERIMENTS

The toy experiments for both the one-layer RADD and GPT-2 models were conducted using the hyperparameters detailed in Table 8. All models were trained for a total of 3,000 steps. In addition to the common parameters, the RADD model utilized an exponential moving average (EMA) with a decay rate of 0.9999.

Table 8: Hyperparameters for toy experiments.

| Hyperparameter | Value |
|---|---|
| Batch Size | 256 |
| Learning Rate | $3 \times 10^{-4}$ |
| Gradient Clipping | 1.0 |
| Weight Decay | 0.0 |
| Dropout | 0.02 |
| Learning Rate Warmup Steps | 1,000 |
| Hidden Dimension | 256 |
| Number of Attention Heads | 1 |

## E.2  SAMPLING STRATEGY IN TOY EXPERIMENTS

In our real-world experiments, the LLaDA model employs a confidence based sampling strategy where the next token to unmask is selected based on confidence scores (Kim et al., 2025). For the controlled toy experiments, however, we adopted a simpler method to ensure a fair comparison between the MDM and ARM. We utilized top-$k$ sampling with $k$=3 for all generations. In this approach, the model restricts its choice to the $k$ most probable tokens from its output distribution and then samples from this reduced set.

The implementation of top-$k$ sampling differs slightly based on the model architecture. For the ARM (GPT-2), given a prompt, the model computes a probability distribution for the next token in the sequence. It then samples from the top $k$ candidates to continue the generation. For the MDM (RADD), the process is applied to the masked position. The model computes a probability distribution over the vocabulary for the [**M**] token and samples from the top $k$ choices to fill that position.

This consistent sampling strategy allows for a direct and fair evaluation of each model's capabilities on the toy tasks.

# F    DETAILS ON ATTENTION ANALYSIS

## F.1    METHODOLOGY FOR PERMUTATION-BASED ANALYSIS

This section details the methodology used in analyzing the attention correlation and the attention weight dynamics in Section 4.3 for each sequence length $L = 10, 20, 30, 40$.

To obtain the correlation as a function of $\Delta_1 + \Delta_2$ and attention weights, we measured across all corresponding pairs of forward and reverse positional permutations. A forward permutation refers to a unique placement of the character pair where the lowercase letter precedes the uppercase one. For instance, in a sequence of length $L = 4$, the set of forward permutations and their corresponding relative distances ($\Delta_1$) are:

$$\texttt{aA00}\ (\Delta_1 = 1),\ \texttt{a0A0}\ (\Delta_1 = 2),\ \texttt{a00A}\ (\Delta_1 = 3),$$
$$\texttt{0aA0}\ (\Delta_1 = 1),\ \texttt{0a0A}\ (\Delta_1 = 2),\ \texttt{00aA}\ (\Delta_1 = 1),$$

$$\cdots$$

$$\texttt{zZ00}\ (\Delta_1 = 1),\ \texttt{z0Z0}\ (\Delta_1 = 2),\ \texttt{z00Z}\ (\Delta_1 = 3),$$
$$\texttt{0zZ0}\ (\Delta_1 = 1),\ \texttt{0z0Z}\ (\Delta_1 = 2),\ \texttt{00zZ}\ (\Delta_1 = 1)$$

A reverse permutation is one where the uppercase letter precedes the lowercase one. The corresponding reverse permutations for the examples above have the same relative distances ($\Delta_2$):

$$\texttt{Aa00}\ (\Delta_2 = 1),\ \texttt{A0a0}\ (\Delta_2 = 2),\ \texttt{A00a}\ (\Delta_2 = 3),$$
$$\texttt{0Aa0}\ (\Delta_2 = 1),\ \texttt{0A0a}\ (\Delta_2 = 2),\ \texttt{00Aa}\ (\Delta_2 = 1),$$

$$\cdots$$

$$\texttt{Zz00}\ (\Delta_2 = 1),\ \texttt{Z0z0}\ (\Delta_2 = 2),\ \texttt{Z00z}\ (\Delta_2 = 3),$$
$$\texttt{0Zz0}\ (\Delta_2 = 1),\ \texttt{0Z0z}\ (\Delta_2 = 2),\ \texttt{00Zz}\ (\Delta_2 = 1)$$

For each corresponding pair of forward and reverse permutations with distances $\Delta_1$ and $\Delta_2$, the correlation between their respective attention scores is calculated. To obtain these scores for the analysis, the lowercase character in each permutation is replaced with a $[\mathbf{M}]$ token. We then measure the attention the $[\mathbf{M}]$ token pays to the uppercase context character ($\texttt{A}\ldots\texttt{Z}$).

The analysis for both the attention score correlation and the attention weight dynamics follows this identical permutation-based averaging procedure. The only distinction lies in the specific quantity measured: the former uses the raw dot-product attention scores ($\mathbf{q}^\top R(\Delta)\mathbf{k}$), while the latter uses the softmax-normalized attention weights. This approach ensures that our findings reflect the fundamental behavior of the attention mechanism, independent of specific token positions.

## F.2    FURTHER ATTENTION ANALYSIS RESULTS

This section presents the full results of our attention analysis for all tested sequence lengths, complementing the findings discussed in Section 4.3.

Fig. 9 (left column) shows the empirical correlation of attention scores as a function of total relative distance, $\Delta_1 + \Delta_2$, for sequence lengths $L = 10, 20, 30$, and $40$. The results across all four settings suggest that the correlation between forward and reverse attention scores trends positive, despite considerable variance in the measurements. This consistent positive trend across different sequence lengths suggests that the underlying architectural property is a robust feature of the model, not an artifact of a specific configuration.

Similarly, Fig. 9 (right column) visualizes the dynamics of the softmaxed attention weights for both forward and reverse contexts throughout the 3,000 training steps. In all tested sequence lengths, the weights for both contexts demonstrate a strong parallel trajectory, rising sharply and converging together in a coordinated pattern. This co-movement provides strong visual evidence that the

model establishes the association of both directions while learning with a single direction train data, reinforcing our claim that this behavior is driven by the underlying correlation induced by the full-attention mechanism.

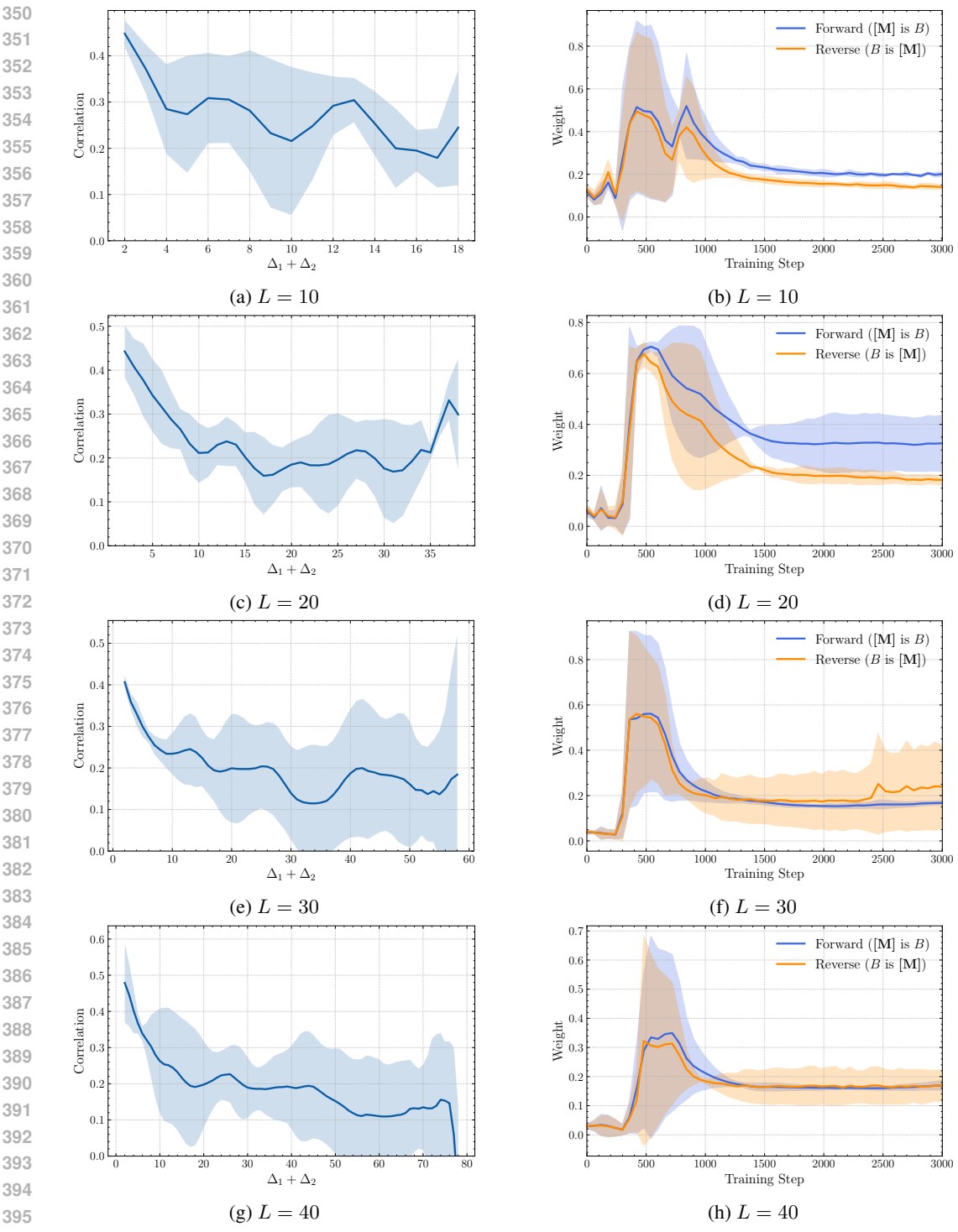

Figure 9: Empirical validation of the attention mechanism's role in reverse inference across various sequence lengths. The left column shows the correlation of attention scores as a function of total relative distance $\Delta_1 + \Delta_2$, while the right column shows the dynamics of softmaxed attention weights for forward (blue) and reverse (orange) contexts during training. Each row corresponds to a different $L = 10, 20, 30, 40$, respectively. Across all settings, the plots reveal two key findings: (1) a consistent positive correlation between forward and reverse attention scores, and (2) a strong parallel trajectory in the development of attention weights. Taken together, these results provide strong empirical evidence that the full-attention architecture inherently couples forward and reverse contexts, driving the concurrent learning of both directions.

## F.3 DYNAMICS OF ATTENTION SCORES AND WEIGHTS

To further investigate the training dynamics, we analyze the raw attention scores (pre-softmax logits) in Fig. 10. This analysis explains the decreasing behavior of the attention weights observed in Fig. 6b.

The model first learns to attend to $B$ while recovering $A$ from "[**M**] is $B$". This corresponds to the period where the model learns to look up $B$ to reconstruct $A$. During this period, the attention to the mask token itself ([**M**] → [**M**]) is suppressed, dropping significantly.

In the later phase, the [**M**] → $B$ score saturates and remains stable. However, a secondary dynamic emerges: the [**M**] → [**M**] attention score begins to recover and rise, similar to model attending special tokens observed in Clark et al. (2019).

This behavior provides a mechanical explanation for the plateau observed in the Fig. 6b. The softmax function computes the weight for token $j$ as $w_j = \exp(s_j)/\sum_k \exp(s_k)$. Even if the score for the informative token $B$ ($s_B$) remains constant, the rising score for the self-token $M$ ($s_M$) increases the denominator of the softmax function. Consequently, the increasing competition from the self-attention score naturally causes the normalized weight assigned to $B$ to decrease, despite $B$ remaining the dominant context.

Crucially, throughout this process, the forward and reverse dynamics remain synchronized. The distinct evolution of these scores confirms that the model maintains a strong, stable link to the informative context $B$, while the fluctuations in final weights are a byproduct of the evolving self-attention mechanism.

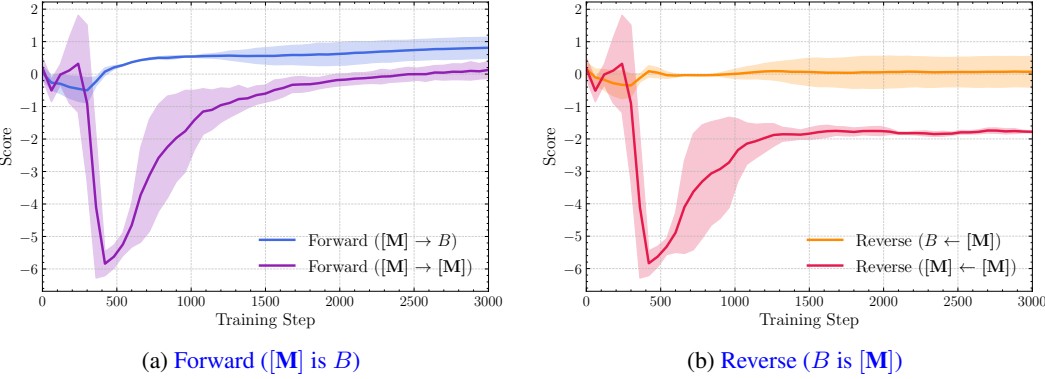

(a) Forward ([**M**] is $B$)  (b) Reverse ($B$ is [**M**])

Figure 10: Training dynamics of raw attention scores in the MDM. (a) Forward scores ([M] → . . . ) for the sequence "[M] is $B$". The score to the informative token $B$ (blue) rises sharply and stabilizes, while the self-attention score to [M] (purple) is initially suppressed before recovering. (b) Reverse scores ( . . . ← [**M**]) for "$B$ is [M]". Crucially, the reverse dynamics mirror the forward ones: attention to $B$ from the masked position [M] (orange) follows a similar trajectory to the forward informative score. The recovery of the score to [M] in both directions increases the softmax denominator, inducing the decrease in attention weights in later training stages (Fig. 6b).

