# OpenReview forum: "Breaking the Reversal Curse: How Masked Diffusion Models Achieve Reverse Inference"
_ICLR.cc/2026/Conference — Submitted to ICLR 2026_

### Official Review · Reviewer_8J1T · 2025-10-29

**Soundness:** 2
**Presentation:** 3
**Contribution:** 2
**Rating:** 2
**Confidence:** 4

**Summary:**

The authors examine the reversal curse: the inability of autoregressive models to reverse the logic of data given at training time. The authors note that traditional wisdom for why MDMs can overcome the reversal curse is faulty; the exact reversal logic is not covered in train time even with arbitrary masking. However, intuitively the masked training example is connected to the exact reversal via global attention scores, and the authors examine some theory for why the connection emerges. These are accompanied by empirical evaluations of large scale MDMs and ARMs on reversal tasks as well as on synthetic toy data.

**Strengths:**

- The authors do point out the misconception that the reversal problem is covered during MDM training due to masking, and correct it by noting that the masking that appears in training is subtly different.

**Weaknesses:**

- The theory result doesn't really provide much insight into what's going on empirically. As the authors remark, the "simplifying assumptions of independent, isotropic queries and keys [do not hold] in practice". While the intuition that [M] is B and B is [M] are linked by correlative attention scores makes sense, it is not clear why, a) the scores correlate without the assumptions and, b) passing through general transformer layers will maintain high probability on A for B is [M].

- A strong standard for "reversal curse" theory is Theorem 3 in [Zhu et. al Towards a Theoretical Understanding of the ‘Reversal
Curse’ via Training Dynamics], which provides a guarantee that after some point in training the reversal probability is essentially uniform. They also avoid making strong assumptions about q/k vectors, so a similar result showing a lower bound on the probability of A for "B is [M]" would be a stronger theory result to have.

- On the experimental side, the toy experiments are insightful, but corresponding analysis for general text MDMs like LLaDa are needed to confirm that these attention correlations are sustained when training on general text datasets.

**Questions:**

- Do the attention score correlations in Figure 6 also appear in the LLaDa model?
- Are there any empirical properties you can observe about the q/k vectors that loosen the strong assumptions of Theorem 4.1 but still make the correlation result hold?
- In the toy example, why do both attention weights increase and then decrease to a plateau during training?

---

> ### Author Response · Authors · 2025-11-20
> **Response to Reviewer 8J1T**
>
> We sincerely thank you for your time and feedback. We appreciate your acknowledgement of our clarification regarding misconceptions about the masking-based explanation for MDMs and your recognition of the empirical value of our toy experiments.
>
> **Response to W1 & W2: On the Theoretical Assumptions and Significance**
>
> In the revised manuscript, we relax the original independence/isotropy assumptions while preserving the key qualitative conclusion that the expected forward-reverse attention correlation is positive, and we additionally provide a Cantelli inequality for this correlation in Appendix C.3. Concretely, we no longer assume independence or restrict the conditional mean of $\mathbf{k} | \mathbf{q}$. instead, we only require that $\mathbf{k} | \mathbf{q}$ respect the 2D rotational structure induced by RoPE. For clarity, the revised theorem focuses on the symmetric case $\Delta_1=\Delta_2$, and we are investigating extensions to the more general case $\Delta_1\neq\Delta_2$. Please see Section 4.2 and Appendix C for the precise assumptions, statements, and proofs.
>
> Although our theoretical analysis does not treat multilayer and is not as strong as the training-dynamics guarantee of Zhu et al. [1], we believe it provides a meaningful first step toward understanding reversal behavior in diffusion-based LLMs. To complement our simplified theory, we added new experiments on the full multilayer LLaDA model (Section 4.4). These results show that, as in the toy setting, LLaDA achieves high reversal accuracy and exhibits a clear positive correlation between forward and reverse attention scores. Figure 7 in the revised manuscript mirrors the patterns observed in Figures 5 and 6-(a) of the toy experiment. Together, these findings give empirical support for our proposed architectural explanation in a realistic large-scale setting.
>
> **Response to W3 & Q1: On the Attention Correlation in Large-Scale Models (LLaDA)**
>
> In response, we have conducted the same attention-score correlation analysis on the full LLaDA-8B model. Figure 7-(c) shows a clear and consistent positive correlation between the forward and reverse attention scores, closely mirroring the behavior observed in the toy experiment (Figure 6-(a)). This confirms that the correlation mechanism is not an artifact of the synthetic setting but also appears when training on general text at scale. The revised manuscript reports this analysis and the corresponding figure in Section 4.4.
>
>
>
> **Response to Q3: On the Dynamics of Attention Weights in the Toy Example**
>
> While we do not fully understand the underlying training dynamics, we can outline our current understanding.
>
> In the early phase of training, the sharp increase in attention to the corresponding token is the correct behavior. The model quickly learns to attend strongly to the token B in “[M] is B,” and this is precisely the key mechanism that enables it to recover A from “B is [M]”.
>
> As training progresses, we observe that the attention scores required for the reversal behavior remain in place. We also find that the attention score to the mask token itself increases, similar to the model attending to special tokens (Clark et al. [2]). As attention grows on the mask token, the weight assigned to the correct token in the softmax-normalized distribution naturally decreases.
>
> To reflect this interpretation, we have added an explanation in Appendix F.3 of the revised paper. Importantly, throughout training, the forward and reverse attention weights evolve together, and this synchronized behavior is the key piece of evidence supporting the correlation mechanism proposed in our analysis.
>
> ---
>
> References
>
> [1] Zhu, H., Huang, B., Zhang, S., Jordan, M., Jiao, J., Tian, Y., and Russell, S., Towards a Theoretical Understanding of the "Reversal Curse" via Training Dynamics. NeurIPS, 2024.
>
> [2] Clark, K., Khandelwal, U., Levy, O., & Manning, C. D.,  What Does BERT Look At? An Analysis of BERT's Self-Attention. ACL Workshop, 2019.

---

### Official Review · Reviewer_vPFL · 2025-11-03

**Soundness:** 3
**Presentation:** 3
**Contribution:** 2
**Rating:** 4
**Confidence:** 4

**Summary:**

This paper studies a phenomenon appearing in Masked Diffusion Models (MDMs) - previous work has observed that such models break the reverse curse faced by autoregressive language models. However, no formal justification were given for this phenomenon and a common hand-wavy argument of "any-order" modeling was often referred to. The authors first show that the "any-order" argument is misleading by pointing out that the two probability distributions used for reverse unmasking and reverse inference are actually different. Then they provide a new theoretical justification based on the attention score correlation, leveraging a property of the ROPE positional encoding. Finally, empirical evidence was provided by plotting the attention score correlation in real problems.

**Strengths:**

* The paper is well-written, has a clear scientific question to ask (why MDMs have better performance than AR models in reverse inference problems), and then provide a theoretical answer together with empirical justification.
* The theory explaining why MDMs break the reverse curse appears sound. I checked the proofs and they look correct under the given assumptions.
* The empirical evidence seems to align well with the theory.
* I would like to also highlight one aspect - this paper is rare in the sense that it combines both statistical and architectural analysis to investigate a real question - most ML papers focus on one side of them. Papers on generative models often spend most main text playing with probability distributions and leave only few words on architecture specifics - while deep learning architecture papers seldom look at the statistical structure of data distributions. And we need more research like this in the community.

**Weaknesses:**

* There are some claims that need a bit more back-ups in the main text. For example, "During training, the [M] token must attend strongly to B, which increases the forward score S_fwd", although I agree this is likely to happen, it would be better to have evidence about making this an assumption.

* The attention score correlation in Figure 6 only shows the MDMs. What about the AR models?

**Questions:**

Please see weaknesses (I will consider increase the score if these results are provided).

---

> ### Author Response · Authors · 2025-11-20
> **Response to Reviewer vPFL**
>
> We sincerely thank the reviewer for the detailed comments and recognizing the positive aspects of this work. We agree that the points you raised are valid.
>
> **Response to W1: On the claim: "[M] token must attend strongly to B"**
>
>
> During training, the model is exposed only to forward pairs $(A_i, B_i)$ in the form “$A_i​$ is $B_i$​.” When $A_i$​ is masked (i.e., “[M] is $B_i$​”), an informative token for recovering $A_i$​ is $B_i​$. Thus, for the model to correctly predict $A_i$, it must place substantial attention on $B_i$. This behavior is well supported by prior work on masked-token prediction in encoder-based models (e.g., Clark et al. [1]) as well as recent analyses showing that models learn to focus attention on informative context tokens in decoder models (Zucchet et al. [2]).
>
> Our empirical results also confirm this. Figure 6-(b) shows that, during the forward phase of training, the attention weight from the [M] token to B becomes consistently strong. Since a large softmaxed attention weight requires the corresponding raw score to be comparatively large, this directly implies that the forward attention score $S_{fwd}$ must also be high. We revised the text to make this connection explicit and added Figure 10 and Appendix F.3 to further empirically demonstrate the attention score.
>
>
> **Response to W2: On the absence of AR models in Figure 6**
>
> We agree that this is the natural question to ask. However, the attention structure in ARMs fundamentally differs from that of MDMs. In ARMs, attention is strictly causal and computed under a next-token prediction objective: in “A is B,” the token A cannot attend to B, which structurally blocks the correlation with “B is A", unlike MDMs. Because this attention pathway does not exist in ARMs, the forward-reverse coupling observable in MDMs (“[M] is B” and “B is [M]”) cannot be meaningfully examined in ARMs using the same correlation analysis.
>
> Existing research (Zhu et al. [3]) have shown the reverse probability provably collapses to a uniform distribution under ARM objective, a failure we reproduced empirically. This confirms that while ARMs are hindered by their causal structure, MDMs’ encoder architecture facilitates the necessary positive forward-reverse coupling. This contrast is precisely the architectural difference our analysis aims to explain.
>
>
> ---
>
> References
>
> [1] Clark, K., Khandelwal, U., Levy, O., & Manning, C. D.,  What Does BERT Look At? An Analysis of BERT's Self-Attention. ACL Workshop, 2019.
>
> [2] Zucchet, N., Bornschein, J., Chan, S. C., Lampinen, A. K., Pascanu, R., & De, S., How do language models learn facts? Dynamics, curricula and hallucinations. COLM 2025.
>
> [3] Zhu, H., Huang, B., Zhang, S., Jordan, M., Jiao, J., Tian, Y., & Russell, S. J., Towards a theoretical understanding of the 'reversal curse' via training dynamics. NeurIPS, 2024.

---

### Official Review · Reviewer_xNBw · 2025-11-04

**Soundness:** 2
**Presentation:** 3
**Contribution:** 2
**Rating:** 4
**Confidence:** 3

**Summary:**

This paper studies the so-called “reversal curse” in LLMs, where GPTs are observed to fail on reverse queries while the alternative masked diffusion models (MDMs) perform significantly better. The authors provide a larger-scale empirical comparisons between AR models (LLaMA-3.1-8B, Qwen-2.5-7B) and a diffusion-based model (LLaDA-8B) and uncover that the advantages of MDMs for this task lie in their encoder architecture. Theoretical analysis is done on a simple one-layer transformer encoder architecture and toy synthetic experiments are conducted to validate their theoretical predictions. The key claim is that positive correlation between forward and reverse attention scores—induced by the encoder’s full-attention structure—allows MDMs to implicitly link p(x=A|y=B) and p(y=A|x=B), enabling reversal even without explicit training on reversed data.

**Strengths:**

The paper is clearly written and the storyline is easy to follow.
The perspective of linking the correlation induced by the encoder architecture to the reversal task is novel to me.
The theoretical analysis and the toy experimental setting are well-motivated and intuitively connected to the main claim.
Moreover, the inclusion of controlled synthetic experiments provides concrete validation of the theory, making the overall argument coherent and convincing.

**Weaknesses:**

### Vague problem formulation
I don't think the reversal problem studied in this work is well-defined. Depending on the specific scenarios, "A is B" may not the same as "B is A". For examples in the paper such as "Nikki Holland’s child is Tom Holland", the reversal problem requires the model to understand the concept of child and parent, which is not discussed or modeled in this work.
Therefore, the designed toy setting and the synthetic toy task seems less relevant to the so called “reversal curse”.
It may be more clear to just studying how LLMs learn the concept of equality, i.e., "A=B" is strictly equivalent to "B=A".


### Limited significance and practical impact

The “reversal curse” itself is a fairly narrow behavioral artifact. Demonstrating that MDMs perform better on this particular test does not necessarily imply a more general cognitive or practical advantage. I think the work is solid within its niche, but the overall importance of the problem is low.
It is unclear how understanding or “solving” the reversal problem translates into improved real-world functionality of LLMs. This paper would be stronger if it connected the analysis to some practical algorithms or architecture modifications for important tasks such as reasoning, compositional generalization, or real-world applications.

**Questions:**

* If we were to stud how LLMs learn the concept of equality, i.e., "A=B" is the same as "B=A", with the extra symbol "=", would the theoretical analysis or empirical results change?

* I am not fully convinced that results shown in Table 1 should sound any alarm to GPTs. I wonder how will stronger GPTs such as ChatGPT handle the tasks evaluated in Table 1 by pure in-context learning with chain of thoughts.

* How would the claimed architectural correlation translate to deeper, multi-layer Transformers with residual connections and multi-head attention? Does the correlation persist or diminish?

---

> ### Author Response · Authors · 2025-11-20
> **Response to Reviewer xNBw**
>
> We thank the reviewer for the detailed comments and time reviewing this work. Let us address the concerns and questions.
>
>
> **Response to W1: On the Problem Formulation and Toy Setting**
>
> We follow the formulation introduced in “The Reversal Curse: LLMs trained on ‘A is B’ fail to learn ‘B is A’” [1], which established standard notation for reversal problems [2, 3, 4]. This formulation is not limited to equality but includes any reversible or invertible relations. So, examples used in this work are not meant to assert equality even though we use the term “A is B” for readability and clarity.
>
> What makes the reversal problem interesting is that LLMs already understand these underlying concepts (e.g., parents vs. child), but still fail to answer the reversed query. This phenomenon where ARMs fail while MDMs partially succeed is the regime we aim to explain.
>
> To investigate reversible relations, we designed implicit relations (lowercase→uppercase). By avoiding explicit symbols, we eliminate semantic cues and data-level hints. This allows us to isolate and examine our theoretical finding that the architecture alone drives success in the unseen reverse direction. Regarding the suggestion to study equality (“A=B”), this is a special case of reversible relations where the forward and reverse operators are identical. As the underlying mechanism is unchanged, the theoretical and empirical findings remain consistent.
>
>
> **Response to W2: On the Significance and Broader Impact**
>
> The underlying issue exposed by the reversal curse is broader and highly relevant to model reliability. The failure of ARMs highlights a fundamental limitation of left-to-right training objectives in capturing bidirectional relational structure, which underlies many reasoning, compositional, and relational tasks. In contrast, MDMs do succeed at a nontrivial portion of these reversed queries, and our work provides the first architectural account explaining why this occurs.
>
> Identifying the architectural origin of this behavior is practically meaningful because it informs the design of models that need to reason bidirectionally or infer symmetric or invertible relations (tasks such as relation extraction, factual QA, entity linking, and knowledge graph completion). Rather than proposing direct architectural modifications, our contribution is to reveal a previously unarticulated mechanism: encoder-based models possess an inherent coupling between forward and reverse inference due to the geometry of RoPE-driven attention, while ARMs do not. Understanding this difference is a necessary first step toward designing models or objectives that mitigate the reversal failure more broadly.
>
>
> **Response to Q1: On using “A = B’’**
>
> As we mentioned above, our theoretical analysis does not impose any assumption about the semantics of equality. Rather Theorem 4.1 concerns how RoPE-based attention couples the forward and reverse configurations at the architectural level. Therefore it continues to apply even though the relation expressed with “=”.
>
>
> **Response to Q2: On whether stronger GPT models or prompting could solve the task**
>
>
>  We agree that more powerful models (especially those with strong reasoning abilities or in-context learning like ChatGPT) may succeed on this task. However, the surprising aspect of the phenomenon is that a very simple and logically invertible task remains unsolved by reasonable ARMs. Our aim is to understand why these standard models fail in such a basic setting, not to claim that larger instruction-tuned models cannot succeed. We believe this question remains interesting and informative.
>
>
> **Response to Q3: On deeper or multi-layer models**
>
> It is a very natural question to ask how far the theory extends to real multi-layer architectures. Although the theorem is stated for a single layer to keep the derivation tractable, our empirical analyses indicate that the core phenomenon continues to appear in deeper MDMs trained on real data. The updated Figure 7-(c) shows the correlation of LLaDA and supports that the correlation persists in deeper, more complicated structures confirming our theory still holds. The revised manuscript describes the analysis in Section 4.4.
>
>
> ---
> References
>
>
> [1] Berglund, L., Tong, M., Kaufmann, M., Balesni, M., Stickland, A. C., Korbak, T., & Evans, O. . The Reversal Curse: LLMs trained on" A is B" fail to learn" B is A". ICLR, 2024.
>
> [2] Lin, Z., Fu, Z., Liu, K., Xie, L., Lin, B., Wang, W., Cai, D., Wu, Y., and Ye, J. Delving into the Reversal Curse: How Far Can Large Language Models Generalize? NeurIPS, 2024.
>
> [3] Golovneva, O., Allen-Zhu, Z., Weston, J. E., & Sukhbaatar, S. Reverse Training to Nurse the Reversal Curse. COLM, 2024.
>
> [4] Lu, Z., Jin, L., Li, P., Tian, Y., Zhang, L., Wang, S., Xu, G., Tian, C., and Cai, X. Rethinking the Reversal Curse of LLMs: A Prescription from Human Knowledge Reversal. EMNLP, 2024.

---

### Official Review · Reviewer_upZK · 2025-11-05

**Soundness:** 3
**Presentation:** 3
**Contribution:** 2
**Rating:** 4
**Confidence:** 3

**Summary:**

The paper studies the 'reversal curse' in left-to-right autoregressive language models (ARMs) where given training data of the form "A is B", the model struggles to learn the reverse direction ("B is A"). Others e.g. (Nei et al. 2025) have shown that masked diffusion models (MDMs) are effective at tackling the reversal curse.

The authors study this here at larger scale showing that MDMs at larger scale (7-8B) outperform ARMs at the 'reversal curse'. The authors also provide a theoretical explanation by showing that under certain assumptions the correlation between the forward and reverse attention scores is lower bounded by a constant > 0.4 and provide further experiments.

**Strengths:**

This is an interesting problem that ARMs struggle with and the authors seek to provide both theoretical and empirical analysis.

**Weaknesses:**

-Its not super clear to me how insightful the theory is. Is it unique to the transformer attention or would similar correlations be present in any neural network and is this just a consequence of shared parameters?

-The paper focuses on MDMs but is the diffusion aspect necessary? It seems that the loss at the bottom of page 2 is similar to the masked language model loss (e.g. BERT) which also uses a transformer architecture.

-Other authors have already empirically shown that MDMs are effective at the reversal curse which reduces the novelty of this paper.

**Questions:**

See questions above.

---

> ### Author Response · Authors · 2025-11-20
> **Response to Reviewer upZK**
>
> We thank the reviewer for the feedback and time reviewing this work.
>
>  **Response to W1: On the Insightfulness and Uniqueness of the Theory**
>
> Thank you for raising this point. Our theoretical result is intended as an explanation tailored to the specific architecture used in MDMs, not as a universal property of all neural networks. The positive correlation we derive arises from the interaction of shared query-key projections and RoPE-based relative positions in a full-attention Transformer encoder. This mechanism is not present in ARMs and does not automatically follow from parameter sharing alone.
>
> Crucially, analyzing the attention scores themselves reveals how architectural symmetries directly couple the forward and reverse inference paths, providing a mechanistic explanation that cannot be obtained from loss-level or representation-level arguments alone. This makes the theory insightful because it identifies the exact computational interface dot-product attention under RoPE through which forward training leaves a measurable footprint on unseen reverse queries.
>
> Our goal was to explain the observed empirical discrepancy between MDMs and ARMs on reversal tasks, and the theory is designed specifically for that comparison. While the result may extend to other architectures with similar properties, establishing such generality is beyond the scope of the current work.
>
> **Response to W2: On the Necessity of Diffusion (vs. BERT/MLM)**
>
> The reversal curse has been documented mostly in generative LLMs, particularly ARMs, where left-to-right decoding creates a directional bias that hinders reverse inference.. Masked Diffusion Models (MDMs) are one of the first scalable generative alternatives to ARMs, so our goal was to explain why these two generative paradigms behave differently on the same phenomenon.
>
> That said, your intuition is essentially correct: BERT-style Masked Language Models (MLMs), which also use a Transformer encoder and a masked-token prediction objective, would likely exhibit similar reversal behavior. Our architectural explanation extends naturally to them. We likewise strongly agree that BERT should be capable of performing the reversal task as well. However, because BERT is not a generative model, we limit our experiments to generative models where the problem is well established.
>
>
> **Response to W3: On the Novelty of the Paper**
>
> While prior work has observed that MDMs perform well on reversal tasks, our contribution is not the observation itself. The novelty lies in two parts:
>
> 1. Systematic large-scale evaluation:\
>  We provide the first controlled comparison between ARMs and MDMs at the 7-8B scale, using consistent forward, reverse protocols. Prior work evaluated only small models and lacked such systematic analysis.
>
>
> 2. Theoretical explanation:\
> We clarify that prior work’s training-based explanation for why MDMs succeed at reversal is incomplete and introduce the first architectural account of this phenomenon, showing that encoder-based attention induces a positive correlation between forward and reverse attention scores, a mechanism absent in ARMs. We additionally validated our theoretical findings in both toy settings and real-world LLM experiments.
>
> Taken together, our work moves beyond documenting the effect and provides both extensive evaluation and a rigorous explanation of the underlying mechanism.

---

### Author Response · Authors · 2025-11-20
**General Response by Authors**

We sincerely thank all reviewers for their thoughtful and constructive feedback. Across the reviews, we were grateful to see appreciation for the clarity of presentation (xNBw, vPFL); the novelty of investigating reversal behavior through theoretical analysis and controlled empirical experiments (upZK, vPFL, xNBw); combining statistical and architectural analysis(vPFL); clarifying MDM training does not inherently address the reversal problem(8J1T). We appreciate these encouraging assessments .

Reviewers additionally raised several important questions. Regarding those concerns, we address each of them below in detail and update the manuscript with all corresponding changes highlighted in blue.

The major updates are summarized below:


- Additional real-world experiment result in Section 3 and Table 1 (lines 210-231)


- Generalized theorem under more realistic assumptions, along with updated proofs and stronger tail bound inequality in Section 4.2, Appendices C.1 and C.3 (lines 348-377, 810-928, 1026-1089)


- Empirical validation result of Theorem 4.1 at scale in Section 4.4 and Figure 7. (lines 486-523)


- Description of the updated real-world experiment in Appendix D and Table 7 (lines 1093-1098, 1126-1130, 1199-1207)


- Attention score & weight dynamics in Appendix F.3 and Figure 10 (lines 1404-1445)


- Minor typo corrections in captions (Figure 3, 4)

---

### Author Response · Authors · 2025-12-01
**Comment to AC by Authors**

Thank you for taking the time to assess our submission.
We would like to provide a brief, consolidated summary explaining how our rebuttal addresses the reviewers’ major concerns, especially given that only the pre-discussion scores remain visible on your end.


**1. Significance & Scope: Addressing a Fundamental Generative Limit.**

We emphasize that the "reversal curse" is a fundamental structural limitation of Autoregressive Models (ARMs) in logical generalization, not a niche artifact. In contrast, MDMs succeed at a significant portion of reverse inference. Uncovering the architectural mechanism behind this success is crucial because it provides a guide for designing models. Our work focuses on explaining why generative MDMs succeed where generative ARMs fail, establishing a foundational understanding for overcoming the reversal failure in broader generative applications.


**2. Novelty: The First Mechanistic Explanation.**

Prior works have only empirically observed that MDMs outperform ARMs on reversal inference, vaguely attributing this to training objective. Our work is the first to identify and prove the architectural origin (the interaction of RoPE and Bidirectional Attention). We move the field beyond merely documenting the phenomenon to providing a concrete mechanistic explanation of how the architecture inherently couples forward and reverse inference.


**3. Empirical Robustness: Proven at Scale (LLaDA-8B)**

Addressing concerns about the "toy" nature of synthetic experiments, we have added validation on the LLaDA 8B model. Our new results confirm that the attention-score correlation predicted by our theory holds robustly at scale (new Figure 7), effectively closing the gap between our theory and real-world application.


**4. Theoretical Soundness: Generalized Theorem.**

In response to feedback regarding strong assumptions, we have significantly strengthened our main theoretical contribution. We derived a generalized theorem that relaxes the independence and isotropy assumptions. Moreover, we provided a Cantelli-type one-sided bound showing that the correlation remains strictly positive with high probability. This ensures our theoretical claims are rigorous and applicable to realistic Transformer settings.

---

### Meta-Review · Area_Chair_o5ry · 2026-01-04

**Summary:**

This paper aims to explain why Masked Diffusion Models can achieve reverse inference, a task considered to be challenging for LLMs. The papers consists of both theoretical analysis and empirical study. First, the authors conduct systematic evaluation of LLaDA 8B model to show the advantages of MDMs over LLMs on reverse inference. Then, the authors carry out theoretical analysis of this phenomenon in a simplified setting with a one-layer transformer encoder.

**Reviewer Concerns:**

The major criticism from the reviewers includes limited scope, significance, and novelty of the paper, the strong assumptions on the theoretical results, and the gap between theory and experiment results. The authors provide additional experiment results and improved theoretical results with slightly relaxed assumptions. The reviewers are not convinced.

**Reviewer Scores:**

Reviewer vPFL could have increased score slightly

---

### Decision · Program_Chairs · 2026-01-26

Reject